# R-divergence for Estimating Model-oriented Distribution Discrepancy

**Zhilin Zhao**    **Longbing Cao**
Data Science Lab, School of Computing & DataX Research Centre
Macquarie University, Sydney, NSW 2109, Australia
`zhaozhl7@hotmail.com, longbing.cao@mq.edu.au`

## Abstract

Real-life data are often non-IID due to complex distributions and interactions, and the sensitivity to the distribution of samples can differ among learning models. Accordingly, a key question for any supervised or unsupervised model is whether the probability distributions of two given datasets can be considered identical. To address this question, we introduce R-divergence, designed to assess model-oriented distribution discrepancies. The core insight is that two distributions are likely identical if their optimal hypothesis yields the same expected risk for each distribution. To estimate the distribution discrepancy between two datasets, R-divergence learns a minimum hypothesis on the mixed data and then gauges the empirical risk difference between them. We evaluate the test power across various unsupervised and supervised tasks and find that R-divergence achieves state-of-the-art performance. To demonstrate the practicality of R-divergence, we employ R-divergence to train robust neural networks on samples with noisy labels.

## 1  Introduction

Most machine learning methods rely on the basic independent and identically distributed (IID) assumption [7], implying that variables are drawn independently from the same distribution. However, real-life data and machine learning models often deviate from this IID assumption due to various complexities, such as mixture distributions and interactions [9]. For a given supervised or unsupervised task [45], a learning model derives an optimal hypothesis from a hypothesis space by optimizing the corresponding expected risk [53]. However, this optimal hypothesis may not be valid if the distributions of training and test samples differ [54], resulting in distributional vulnerability [63]. In fact, an increasing body of research addresses complex non-IID scenarios [8], including open-world problems [6], out-of-distribution detection [23, 63], domain adaptation [28], and learning with noisy labels [46]. These scenarios raise a crucial yet challenging question: *how to assess the discrepancy between two probability distributions for a specific learning model?*

Empirically, the discrepancy between two probability distributions can be assessed by the divergence between their respective datasets. However, the actual underlying distributions are often unknown, with only a limited number of samples available for observation. Whether the samples from two datasets come from the same distribution is contingent upon the specific learning model being used. This is because different learning models possess unique hypothesis spaces, loss functions, target functions, and optimization processes, which result in varying sensitivities to distribution discrepancies. For instance, in binary classification [26], samples from two datasets might be treated as positive and negative, respectively, indicating that they stem from different distributions. Conversely, a model designed to focus on invariant features is more likely to treat all samples as if they are drawn from the same distribution. For example, in one-class classification [52], all samples are considered positive, even though they may come from different components of a mixture distribution.

37th Conference on Neural Information Processing Systems (NeurIPS 2023).

In cases where the samples from each dataset originate from multiple distributions, they can be viewed as drawn from a complex mixture distribution. We can then estimate the discrepancy between these two complex mixture distributions. This suggests that, in practice, it is both meaningful and necessary to evaluate the distribution discrepancy for a specific learning model, addressing the issue of *model-oriented distribution discrepancy evaluation*.

Estimating the discrepancy between two probability distributions has been a foundational and challenging issue. Various metrics for this task have been proposed, including F-divergence [12], integral probability metrics (IPM)[2], and H-divergence[62]. F-divergence metrics, like Kullback-Leibler (KL) divergence [13] and Jensen Shannon divergence [47], assume that two distributions are identical if they possess the same likelihood at every point. Importantly, F-divergence is not tied to a specific learning model, as it directly measures discrepancy by computing the statistical distance between the two distributions. IPM metrics, including the Wasserstein distance [48], maximum mean discrepancy (MMD)[25], and L-divergence[5, 31], assess discrepancy based on function outputs. They postulate that any function should yield the same expectation under both distributions if they are indeed identical, and vice versa. In this vein, L-divergence explores the hypothesis space for a given model and regards the largest gap in expected risks between the two distributions as their discrepancy. This necessitates evaluating all hypotheses within the hypothesis space, a task which can be computationally demanding. Moreover, hypotheses unrelated to the two distributions could result in inaccurate or biased evaluations. On the other hand, H-divergence [62] contends that two distributions are distinct if the optimal decision loss is greater when computed on their mixed distribution compared to each individual one. It calculates this optimal decision loss in terms of training loss, meaning it trains a minimal hypothesis and evaluates its empirical risk on the identical dataset. A network with limited training data is susceptible to overfitting [50], which can lead to underestimated discrepancies when using H-divergence, even if the two distributions are significantly different.

To effectively gauge the model-oriented discrepancy between two probability distributions, we introduce R-divergence. This novel metric is built on the notion that *two distributions are identical if the optimal hypothesis for their mixture distribution yields the same expected risk on each individual distribution*. In line with this concept, R-divergence evaluates the discrepancy by employing an empirical estimator tailored to the given datasets and learning model. Initially, a minimum hypothesis is derived from the mixed data of the two datasets to approximate the optimal hypothesis. Subsequently, empirical risks are computed by applying this minimum hypothesis to each of the two individual datasets. Finally, the difference in these empirical risks serves as the empirical estimator for the discrepancy between the two probability distributions. The framework of R-divergence is illustrated in Figure 1.

## 2 Model-oriented Two-sample Test

We measure the discrepancy between two probability distributions $p$ and $q$ for a specific supervised or unsupervised learning method. Let $\mathcal{X}$ and $\mathcal{Y}$ be the spaces of inputs and labels, respectively. Assume we observe two sets of $N$ IID samples from the two distributions, i.e., $\widehat{p} = \{x_i\}_{i=1}^{N} \sim p$, and $\widehat{q} = \{x_i'\}_{i=1}^{N} \sim q$, respectively. We assume the mixture distribution $u = (p+q)/2$ and its corresponding mixed data $\widehat{u} = \widehat{p} \cup \widehat{q}$.

For a learning model $\mathcal{T}$, we assume $\mathcal{H}$ is its hypothesis space and $l$ is its $L$-Lipschitz bounded loss function, i.e., $|l(\cdot, \cdot)| \leq c$. For an input $x \in \mathcal{X}$, we suppose $\|x\| \leq B$ and $a \in \mathcal{A}$ is its corresponding target function for the model. For example, $a(x) = y \in \mathcal{Y}$ for supervised classification and $a(x) = x \in \mathcal{X}$ for unsupervised input reconstruction. In general, a loss function can be defined as $l(h(x), a(x))$ for any hypothesis $h \in \mathcal{H}$ and input $x \in \mathcal{X}$. $h(x)$ represents the hypothesis output for $x$. For example, $h(x)$ produces a predicted label by a classifier or the reconstructed input by unsupervised input reconstruction. For the mixture distribution $u$ and its corresponding dataset $\widehat{u}$, we define the expected risk $\epsilon_u(h)$ and empirical risk $\widehat{\epsilon}_{\widehat{u}}(h)$ for any $h \in \mathcal{H}$ [61]:

$$\epsilon_u(h) = \mathbb{E}_{x \sim u} l(h(x), a(x)), \widehat{\epsilon}_{\widehat{u}}(h) = \frac{1}{2N} \sum_{x \in \widehat{u}} l(h(x), a(x)). \tag{1}$$

Then, we define the optimal hypothesis $h_u^*$ and the minimum hypothesis $\widehat{h}_u$, respectively,

$$h_u^* \in \arg\min_{h \in \mathcal{H}} \epsilon_u(h), \widehat{h}_u \in \arg\min_{h \in \mathcal{H}} \widehat{\epsilon}_{\widehat{u}}(h). \tag{2}$$

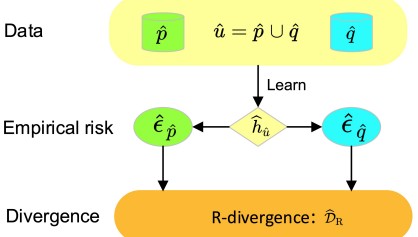

Figure 1: R-divergence estimates the distribution discrepancy on datasets $\widehat{p}$ and $\widehat{q}$. $\widehat{h}_{\widehat{u}}$ represents the minimum hypothesis learned on the mixed data $\widehat{u} = \widehat{p} \cup \widehat{q}$. $\widehat{\epsilon}_{\widehat{p}}$ and $\widehat{\epsilon}_{\widehat{q}}$ represent the empirical risks of the minimum hypothesis on data $\widehat{p}$ and $\widehat{q}$, respectively. R-divergence $\widehat{\mathcal{D}}_{\mathrm{R}}$ estimates the distribution discrepancy by the empirical risk difference between the two datasets.

A learning model of the task aims to learn the minimum hypothesis $\widehat{h}$ on observed data to approximate the optimal hypothesis $h^*$. This aim can be converted to the *model-oriented two-sample test*: For datasets $\widehat{p}$ and $\widehat{q}$ and their learning task with hypothesis space $\mathcal{H}$ and loss function $l$, can we determine whether their probability distributions are identical (e.g., $p = q$) for the specific model? This research question is equivalent to: whether the samples of the two datasets can be treated as sampled from the same distribution for the model?

## 3   R-divergence for Distribution Discrepancy

To fulfill the model-oriented two-sample test, we develop R-divergence to estimate the discrepancy between two probability distributions $p$ and $q$ for a model $\mathcal{T}$. R-divergence regards two distributions as identical if the optimal hypothesis on their mixture distribution has the same expected risk on each individual distribution. Accordingly, with the corresponding datasets $\widehat{p}$ and $\widehat{q}$, R-divergence estimates the optimal hypothesis $h_u^*$ on the mixture distribution $u$ by learning the minimum hypothesis $\widehat{h}_{\widehat{u}}$ on the mixed dataset $\widehat{u}$. Then, R-divergence estimates the expected risks $\epsilon_p(h_u^*)$ and $\epsilon_q(h_u^*)$ on the two distributions by evaluating the empirical risks $\widehat{\epsilon}_{\widehat{p}}(\widehat{h}_{\widehat{u}})$ and $\widehat{\epsilon}_{\widehat{q}}(\widehat{h}_{\widehat{u}})$ on their data samples. R-divergence measures the empirical risk difference between $\widehat{\epsilon}_{\widehat{p}}(\widehat{h}_{\widehat{u}})$ and $\widehat{\epsilon}_{\widehat{q}}(\widehat{h}_{\widehat{u}})$ to estimate the distribution discrepancy.

Accordingly, for a supervised or unsupervised model with hypothesis space $\mathcal{H}$ and loss function $l$, R-divergence evaluates the discrepancy between distributions $p$ and $q$ by,

$$\mathcal{D}_{\mathrm{R}}(p\|q) = d(\epsilon_p(h_u^*), \epsilon_q(h_u^*)) = |\epsilon_p(h_u^*) - \epsilon_q(h_u^*)|. \tag{3}$$

$d(\cdot, \cdot)$ represents the absolute difference. Intuitively, if $p = q$, then $u = p = q$ and $h_u^*$ is the optimal hypothesis for both $p$ and $q$. Thus, the expected risks on the two distributions are the same, which leads to a zero discrepancy. In contrast, if $p \neq q$, $h_u^*$ is the optimal hypothesis for $u$, and it performs differently on $p$ and $q$. Then, the expected risk difference can evaluate the discrepancy between the two distributions for the specific model. Different from F-divergence without considering specific learning models, R-divergence involves the property of the specific model in evaluating the model-oriented distribution discrepancy conditional on the optimal hypothesis. Specifically, R-divergence infers the discrepancy by comparing the distance between the learning results on two distributions. The learning result is sensitive to the learning model, i.e., its hypothesis space and loss function.

Further, $\mathcal{D}_{\mathrm{R}}(p\|q)$ can be estimated on datasets $\widehat{p}$ and $\widehat{q}$. R-divergence estimates the discrepancy by searching a minimum hypothesis $\widehat{h}_u$ on the mixed dataset $\widehat{u} = \widehat{p} \cup \widehat{q}$, and calculates the empirical risks $\widehat{\epsilon}_{\widehat{p}}(\widehat{h}_u)$ and $\widehat{\epsilon}_{\widehat{q}}(\widehat{h}_u)$ on each individual dataset. Then, $\mathcal{D}_{\mathrm{R}}(p\|q)$ can be estimated by the empirical risk difference $\widehat{\mathcal{D}}_{\mathrm{R}}(p\|q)$:

$$\widehat{\mathcal{D}}_{\mathrm{R}}(\widehat{p}\|\widehat{q}) = d(\widehat{\epsilon}_{\widehat{p}}(\widehat{h}_{\widehat{u}}), \widehat{\epsilon}_{\widehat{q}}(\widehat{h}_{\widehat{u}})) = |\widehat{\epsilon}_{\widehat{p}}(\widehat{h}_{\widehat{u}}) - \widehat{\epsilon}_{\widehat{q}}(\widehat{h}_{\widehat{u}})|. \tag{4}$$

$\widehat{\mathcal{D}}_{\mathrm{R}}(\widehat{p}\|\widehat{q})$ can be treated as an empirical estimator for the distribution discrepancy $\mathcal{D}_{\mathrm{R}}(p\|q)$.

Because $\widehat{h}_u$ is the minimum hypothesis on the mixed dataset $\widehat{u}$, $\widehat{h}_u$ will overfit neither $\widehat{p}$ nor $\widehat{q}$. Specifically, both $\widehat{\epsilon}_{\widehat{p}}(\widehat{h}_u)$ and $\widehat{\epsilon}_{\widehat{q}}(\widehat{h}_u)$ will not tend towards zero when $\widehat{p}$ and $\widehat{q}$ are different. This property ensures that the empirical estimator will not equal zero for different given datasets. Further, as $\widehat{u}$ consists of $\widehat{p}$ and $\widehat{q}$, both $\widehat{\epsilon}_{\widehat{p}}(\widehat{h}_u)$ and $\widehat{\epsilon}_{\widehat{q}}(\widehat{h}_u)$ will not tend towards extremely large values, which ensures a stable empirical estimator. In general, due to the property that the minimum hypothesis learned on the mixed data is applied to evaluate the empirical risk on each individual dataset, R-divergence can address the overfitting issue in H-divergence [62]. The procedure of estimating the model-oriented discrepancy between two probability distributions is summarized in Algorithm 1 (see Appendix B.1).

R-divergence diverges substantially from L-divergence [31] and H-divergence [62]. While R-divergences and L-divergences apply to different hypotheses and scenarios, R-divergences and H-divergences differ in their assumptions on discrepancy evaluation and risk calculation. Notably, R-divergence enhances performance by mitigating the overfitting issues seen with H-divergence. H-divergence assumes that two distributions are different if the optimal decision loss is higher on their mixture than on individual distributions. As a result, H-divergence calculates low empirical risks for different datasets, leading to underestimated discrepancies. In contrast, R-divergence tackles this by assuming that two distributions are likely identical if the optimal hypothesis yields the same expected risk on each. Thus, R-divergence focuses on a minimum hypothesis based on mixed data and assesses its empirical risks on the two individual datasets. It treats the empirical risk gap as the discrepancy, ensuring both small and large gaps for similar and different datasets, respectively. Technically, R-divergence looks at a minimum hypothesis using mixed data, while L-divergence explores the entire hypothesis space. Consequently, R-divergence is model-oriented, accounting for their hypothesis space, loss function, target function, and optimization process.

Accordingly, there is no need to select hypothesis spaces and loss functions for R-divergence, as it estimates the discrepancy tailored to a specific learning task. Hence, it is reasonable to anticipate different results for varying learning tasks. Whether samples from two datasets are drawn from the same distribution is dependent on the particular learning model in use, and different models exhibit varying sensitivities to distribution discrepancies. Therefore, the distributions of two given datasets may be deemed identical for some models, yet significantly different for others.

## 4 Theoretical Analysis

Here, we present the convergence results of the proposed R-divergence method, i.e., the bound on the difference between the empirical estimator $\widehat{\mathcal{D}}_R(\widehat{p}\|\widehat{q})$ and the discrepancy $\mathcal{D}_R(p\|q)$. We define the L-divergence [35] $\mathcal{D}_L(p\|q)$ as

$$\mathcal{D}_L(p\|q) = \sup_{h \in \mathcal{H}}(|\epsilon_p(h) - \epsilon_q(h)|). \tag{5}$$

Further, we define the Rademacher complexity $\mathcal{R}_{\mathcal{H}}^l(\widehat{u})$ [4] as

$$\mathcal{R}_{\mathcal{H}}^l(\widehat{u}) \equiv \frac{1}{2N}\mathbb{E}_{\boldsymbol{\sigma} \sim \{\pm 1\}^{2N}}\left[\sup_{h \in \mathcal{H}}\sum_{x \in \widehat{u}}\sigma l(h(x), a(x))\right]. \tag{6}$$

According to the generalization bounds in terms of L-divergence and Rademacher complexity, we can bound the difference between $\widehat{\mathcal{D}}_R(\widehat{p}\|\widehat{q})$ and $\mathcal{D}_R(p\|q)$.

**Theorem 4.1.** *For any $x \in \mathcal{X}$, $h \in \mathcal{H}$ and $a \in \mathcal{A}$, assume we have $|l(h(x), a(x))| \leq c$. Then, with probability at least $1 - \delta$, we can derive the following general bound,*

$$|\mathcal{D}_R(p\|q) - \widehat{\mathcal{D}}_R(\widehat{p}\|\widehat{q})| \leq 2\mathcal{D}_L(p\|q) + 2\mathcal{R}_{\mathcal{H}}^l(\widehat{p}) + 2\mathcal{R}_{\mathcal{H}}^l(\widehat{q}) + 12c\sqrt{\frac{\ln(8/\delta)}{N}}.$$

**Corollary 4.2.** *Following the conditions of Theorem 4.1, the upper bound of $\sqrt{\mathrm{Var}\left[\widehat{\mathcal{D}}_R(\widehat{p}\|\widehat{q})\right]}$ is*

$$\frac{34c}{\sqrt{N}} + \sqrt{2\pi}\left(2\mathcal{D}_L(p\|q) + 2\mathcal{R}_{\mathcal{H}}^l(\widehat{p}) + 2\mathcal{R}_{\mathcal{H}}^l(\widehat{q})\right).$$

The detailed proof is given in Section A. Note that both the convergence and variance of $\widehat{\mathcal{D}}_R(\widehat{p}\|\widehat{q})$ depend on the L-divergence, Rademacher complexity, and the number of samples from each dataset. Recall that L-divergence is based on the whole hypothesis space, but R-divergence is based on an optimal hypothesis. The above results reveal the relations between these two measures: a smaller L-divergence between two probability distributions leads to a tighter bound on R-divergence. As the Rademacher complexity measures the hypothesis complexity, we specifically consider the hypothesis complexity of deep neural networks (DNNs). According to Talagrand's contraction lemma [44] and the sample complexity of DNNs [18, 59], we can obtain the following bound for DNNs.

**Proposition 4.3.** *Based on the conditions of Theorem 4.1, we assume $\mathcal{H}$ is the class of real-valued networks of depth $D$ over the domain $\mathcal{X}$. Let the Frobenius norm of the weight matrices be at most $M_1, \ldots, M_D$, the activation function be 1-Lipschitz, positive-homogeneous and applied element-wise (such as the ReLU). Then, with probability at least $1 - \delta$, we have,*

$$|\mathcal{D}_R(p\|q) - \widehat{\mathcal{D}}_R(\widehat{p}\|\widehat{q})| \leq 2\mathcal{D}_L(p\|q) + \frac{4LB(\sqrt{2D\ln 2} + 1)\prod_{i=1}^{D} M_i}{\sqrt{N}} + 12c\sqrt{\frac{\ln(8/\delta)}{N}}.$$

Accordingly, we can obtain a tighter bound if the L-divergence $D_L(p\|q)$ meets a certain condition.

**Corollary 4.4.** *Following the conditions of Proposition 4.3, with probability at least $1 - \delta$, we have the following conditional bounds if $\mathcal{D}_L(p\|q) \leq \left|\epsilon_u(h_u^*) - \epsilon_u(\widehat{h}_u)\right|$,*

$$|\mathcal{D}_R(p\|q) - \widehat{\mathcal{D}}_R(\widehat{p}\|\widehat{q})| \leq \frac{8LB(\sqrt{2D\ln 2} + 1)\prod_{i=1}^{D} M_i}{\sqrt{N}} + 22c\sqrt{\frac{\ln(16/\delta)}{N}}.$$

Then, the empirical estimator $\widehat{\mathcal{D}}_R(\widehat{p}\|\widehat{q})$ converge uniformly to the discrepancy $\mathcal{D}_R(p\|q)$ as the number of samples $N$ increases. Then, based on the result of Proposition 4.3, we can obtain a general upper bound of $\widehat{\mathcal{D}}_R(\widehat{p}\|\widehat{q})$.

**Corollary 4.5.** *Following the conditions of Proposition 4.3, as $N \to \infty$, we have,*

$$\widehat{\mathcal{D}}_R(\widehat{p}\|\widehat{q}) \leq 2\mathcal{D}_L(p\|q) + \mathcal{D}_R(p\|q).$$

The empirical estimator will converge to the sum of L-divergence $\mathcal{D}_L(p\|q)$ and R-divergence $\mathcal{D}_R(p\|q)$ as the sample size grows, as indicated in Corollary 4.5. As highlighted by Theorem 4.1, a large L-divergence can influence the convergence of the empirical estimator. However, a large L-divergence is also useful for determining whether samples from two datasets are drawn from different distributions, as it signals significant differences between the two. Therefore, both L-divergence and R-divergence quantify the discrepancy between two given probability distributions within a learning model. Specifically, L-divergence measures this discrepancy across the entire hypothesis space, while R-divergence does so in terms of the optimal hypothesis of the mixture distribution. Hence, a large L-divergence implies that the empirical estimator captures a significant difference between the two distributions. Conversely, a small L-divergence suggests that the empirical estimator will quickly converge to a minor discrepancy as the sample size increases, as evidenced in Corollary 4.4. Thus, the empirical estimator also reflects a minimal distribution discrepancy when samples from both datasets are drawn from the same distribution.

## 5 Experiments

We evaluate the discrepancy between two probability distributions w.r.t. R-divergence (R-Div)[1] for both unsupervised and supervised learning models. Furthermore, we illustrate how R-Div is applied to learn robust DNNs from data corrupted by noisy labels. We compare R-Div with seven state-of-the-art methods that estimate the discrepancy between two probability distributions. The details of the comparison methods are presented in Appendix C.

---

[1]The source code is publicly available at: `https://github.com/Lawliet-zzl/R-div`.

## 5.1 Evaluation Metrics

The discrepancy between two probability distributions for a learning model is estimated on their given datasets IID drawn from the distributions, respectively. We decide whether two sets of samples are drawn from the same distribution. A model estimates the discrepancy on their corresponding datasets $\widehat{p} = \{x_i\}_{i=1}^N \sim p$ and $\widehat{q} = \{x_i'\}_{i=1}^N \sim q$. $\widehat{\mathcal{D}}_R(\widehat{p}\|\widehat{q})$ serves as an empirical estimator for the discrepancy $\mathcal{D}_R(p\|q)$ to decide whether $p \neq q$ if its output value exceeds a predefined threshold.

For an algorithm that decides if $p \neq q$, the *Type I error* [14] occurs when it incorrectly decides $p \neq q$, and the corresponding probability of Type I error is named the *significant level*. In contrast, the *Type II error* [14] occurs when it incorrectly decides $p = q$, and the corresponding probability of not making this error is called the *test power* [34]. In the experiments, the samples of two given datasets $\widehat{p}$ and $\widehat{q}$ are drawn from different distributions, i.e., the ground truth is $p \neq q$. Both the significant level and the test power are sensitive to the two probability distributions. Accordingly, to measure the performance of a learning algorithm, we calculate its test power when a certain significant level $\alpha \in [0, 1]$ can be guaranteed. A higher test power indicates better performance when $p \neq q$.

The significant level can be guaranteed with a permutation test [17]. For the given datasets $\widehat{p}$ and $\widehat{q}$, the permutation test uniformly randomly swaps samples between $\widehat{p}$ and $\widehat{q}$ to generate $Z$ dataset pairs $\{(\widehat{p}^z, \widehat{q}^z)\}_{z=1}^Z$. For $\widehat{p}$ and $\widehat{q}$, R-Div outputs the empirical estimator $\widehat{\mathcal{D}}_R(\widehat{p}\|\widehat{q})$. Note that if $p = q$, swapping samples between $\widehat{p}$ and $\widehat{q}$ cannot change the distribution, which indicates that the permutation test can guarantee a certain significant level. If the samples of $\widehat{p}$ and $\widehat{q}$ are drawn from different distributions, the test power is 1 if the algorithm can output $p \neq q$ Otherwise, the test power is 0. Specifically, to guarantee the $\alpha$ significant level, the algorithm output $p \neq q$ if $\widehat{\mathcal{D}}_R(\widehat{p}\|\widehat{q})$ is in the top $\alpha$-quantile among $\mathcal{G} = \{\widehat{\mathcal{D}}_R(\widehat{p}^z\|\widehat{q}^z)\}_{z=1}^Z$. We perform the test for $K$ times using different randomly-selected dataset pairs $(\widehat{p}, \widehat{q})$ and average the output values as the test power of the algorithm. If not specified, we set $\alpha = 0.05$, $Z = 100$ and $K = 100$. The procedure of calculating the average test power of R-Div is summarized in Algorithm 2 (see Appendix B.2).

## 5.2 Unsupervised Learning Tasks

**Benchmark Datasets:** Following the experimental setups as [41] and [62], we adopt four benchmark datasets, namely Blob, HDGM, HIGGS, and MNIST. We consider learning unsupervised models on these datasets, and the considered hypothesis spaces are for generative models. Specifically, MNIST adopts the variational autoencoders (VAE) [33], while the other low dimensional data adopt kernel density estimation (KDE) [27]. For fair comparisons, we follow the same setup as [62] for all methods. Because all the compared methods have hyper-parameters, we evenly split each dataset into a training set and a validation set to tune hyper-parameters and compute the final test power, respectively. The average test powers for a certain significant level on HDGM [41], Blob [41], MNIST [38] and HIGGS [1] are presented in Figure 3 , Figure 4 , Table 8 (see Appendix D.1) and Table 2, respectively. Overall, our proposed R-Div achieves the best test power on all datasets. On HDGM, R-Div obtains the best performance for different dimensions and achieves the perfect test power with fewer samples. Specifically, the test power of R-Div decreases gracefully as the dimension increases on HDGM. On Blob with different significant levels, R-Div achieves the same test power as the second-best method with a lower variance. On MNIST, both H-Div and R-Div achieve the perfect test power on all the sample sizes. On HIGGS, R-Div achieves the perfect test power even on the dataset with the smallest scale.

The discrepancy as measured by R-Div is distinct for two different distributions, even with a small sample size. This advantage stems from mitigating overfitting of H-Div by learning the minimum hypothesis and evaluating its empirical risks across varying datasets. R-Div learns a minimum hypothesis based on the mixed data and yields consistent empirical risks for both datasets when samples are sourced from the same distribution. Conversely, if samples from the two datasets come from different distributions, the empirical risks differ. This is because the training dataset for the minimum hypothesis does not precisely match either of the individual datasets.

To better demonstrate the advantages of R-Div, we measure the amount of overfitting by calculating empirical risks on $p$ and $q$. The results presented in Table 1 indicate that R-Div achieves a notably clearer discrepancy, attributable to the significant differences in empirical risks on $p$ and $q$. This is because R-Div mitigates overfitting by optimizing a minimum hypothesis on a mixed dataset

Table 1: The empirical risks on $p$ and $q$ for L-Div, H-Div, and R-Div on MNIST when $N = 200$.

| Methods | Hypothesis | $\epsilon_p(h_p)$ | $\epsilon_q(h_q)$ | $|\epsilon_p(h_p) - \epsilon_q(h_q)|$ |
|---------|-----------|-------------------|-------------------|----------------------------------------|
| L-Div | $h_p = h_q \in \mathcal{H}$ | 713.0658 | 713.1922 | 0.1264 |
| H-Div | $h_p \in \arg\min_{h \in \mathcal{H}} \epsilon_p(h)$ 
 $h_q \in \arg\min_{h \in \mathcal{H}} \epsilon_q(h)$ | 111.7207 | 118.9020 | 7.1813 |
| R-Div | $h_p = h_q \in \arg\min_{h \in \mathcal{H}} \epsilon_{\frac{p+q}{2}}(h)$ | 144.0869 | 99.6166 | **44.4703** |

Table 2: The average test power $\pm$ standard error at the significant level $\alpha = 0.05$ on HIGGS. $N$ represents the number of samples in each given dataset, and boldface values represent the relatively better discrepancy estimation.

| N | 1000 | 2000 | 3000 | 5000 | 8000 | 10000 | Avg. |
|---|------|------|------|------|------|-------|------|
| ME | 0.120±0.007 | 0.165±0.019 | 0.197±0.012 | 0.410±0.041 | 0.691±0.691 | 0.786±0.041 | 0.395 |
| SCF | 0.095±0.022 | 0.130±0.026 | 0.142±0.025 | 0.261±0.044 | 0.467±0.038 | 0.603±0.066 | 0.283 |
| C2STS-S | 0.082±0.015 | 0.183±0.032 | 0.257±0.049 | 0.592±0.037 | 0.892±0.029 | 0.974±0.070 | 0.497 |
| C2ST-L | 0.097±0.014 | 0.232±0.017 | 0.399±0.058 | 0.447±0.045 | 0.878±0.020 | 0.985±0.050 | 0.506 |
| MMD-O | 0.132±0.050 | 0.291±0.012 | 0.376±0.022 | 0.659±0.018 | 0.923±0.013 | **1.000±0.000** | 0.564 |
| MMD-D | 0.113±0.013 | 0.304±0.035 | 0.403±0.050 | 0.699±0.047 | 0.952±0.024 | **1.000±0.000** | 0.579 |
| H-Div | 0.240±0.020 | 0.380±0.040 | 0.685±0.015 | 0.930±0.010 | **1.000±0.000** | **1.000±0.000** | 0.847 |
| R-Div | **1.000±0.000** | **1.000±0.000** | **1.000±0.000** | **1.000±0.000** | **1.000±0.000** | **1.000±0.000** | **1.000** |

and assessing its empirical risks across diverse datasets. Consequently, R-Div can yield a more pronounced discrepancy for two markedly different distributions.

## 5.3 Supervised Learning Tasks

**Multi-domain Datasets:** We adopt the PACS dataset [40] consisting of samples from four different domains, including photo (P), art painting (A), cartoon (C), and sketch (S). Following [40] and [57], we use AlexNet [37] as the backbone. Accordingly, samples from different domains can be treated as samples from different distributions. We consider the combinations of two domains and estimate their discrepancies. The results in Figure 5 (see Appendix D.2) represent that the test power of R-Div significantly improves as the number of samples increases. In addition, R-Div obtains the best performance on different data scales. The results in Table 3 show that R-Div achieves the superior test power across all the combinations of two domains. This is because the compared H-Div method suffers from the overfitting issue because it searches for a minimum hypothesis and evaluates its empirical risk on the same dataset. This causes empirical risks on different datasets to be minimal, even if they are significantly different. R-Div improves the performance of estimating the discrepancy by addressing this overfitting issue. Specifically, given two datasets, R-Div searches a minimum hypothesis on their mixed data and evaluates its empirical risks on each individual dataset. R-Div obtains stable empirical risks because the datasets used to search and evaluate the minimum hypothesis are not the same and partially overlap.

**Spurious Correlations Datasets:** We evaluate the probability distribution discrepancy between MNIST and Colored-MNIST (with spurious correlation) [3]. The adopted model for the two datasets is fully-connected and has two hidden layers of 128 ReLU units, and the parameters are learned using Adam [32]. The results in Table 4 reveal that the proposed R-Div achieves the best performance, which indicates that R-Div can explore the spurious correlation features by learning a minimum hypothesis on the mixed datasets.

**Corrupted Datasets:** We assess the probability distribution discrepancy between a given dataset and its corrupted variants. Adhering to the construction methods outlined by Sun et al.[55] and the framework of H-Div[62], we conduct experiments using H-Div and R-Div. We examine partial corruption types across four corruption levels on both CIFAR10 [36] and ImageNet [15], employing ResNet18 [22] as the backbone. Assessing discrepancies in datasets with subtle corruptions is particularly challenging due to the minor variations in their corresponding probability distributions. Therefore, we employ large sample sizes, $N = 5000$ for CIFAR10 and $N = 1000$ for ImageNet,

Table 3: The average test power $\pm$ standard error at the significant level $\alpha = 0.05$ on different domain combinations of PACS. P, A, C and S represent photo, art painting, cartoon, and sketch domains, respectively. The distribution discrepancies for six combinations of two domains are estimated. For example, A + C represents the combination of art painting and cartoon domains. Boldface values represent the relatively better discrepancy estimation.

| Datasets | A+C | A+P | A+S | C+P | C+S | P+S | Ave. |
|---|---|---|---|---|---|---|---|
| H-Div | 0.846±0.118 | 0.668±0.064 | 0.964±0.066 | 0.448±0.135 | 0.482±0.147 | 0.876±0.090 | 0.714 |
| R-Div | **1.000±0.000** | **1.000±0.000** | **1.000±0.000** | **0.527±0.097** | **0.589±0.160** | **0.980±0.040** | **0.849** |

Table 4: The average test power on MNIST and Colored-MNIST at the significant level $\alpha = 0.05$.

| ME | SCF | C2STS-S | C2ST-L | MMD-O | MMD-D | H-Div | R-Div |
|---|---|---|---|---|---|---|---|
| 0.383 | 0.204 | 0.189 | 0.217 | 0.311 | 0.631 | 0.951 | **1.000** |

and observe that methods yield a test power of 0 when the corruption level is mild. Consistently across different corruption types, R-Div demonstrates superior performance at higher corruption levels and outperforms H-Div across all types and levels of corruption. Moreover, our results on ImageNet substantiate that the proposed R-Div effectively estimates distribution discrepancies in large-resolution datasets.

**Advanced Network Architectures:** For supervised learning tasks, R-Div selects the network architecture for the specific model. To verify the effect of R-div, we perform experiments on a wide range of architectures. We compare R-div with H-div because both of them are model-oriented and the other comparison methods have specific network architectures. Specifically, we consider the classification task on CIFAR10 and Corrupted-CIFAR10 and adopt four advanced network architectures, including ResNet18 [22], SENet [24], ConvNeXt [42], ViT [16]. The results are summarized in Table 6. For large-scale datasets (N = 10000), both H-Div and R-Div achieve perfect test power with different network architectures because more samples are applied to estimate the probability distribution discrepancy precisely. However, R-Div has the better test power for more challenging scenarios where the datasets are small (N = 1000, 5000). This may be because R-Div addresses the overfitting issue of H-Div by optimizing the minimum hypothesis and evaluating its empirical risk on different datasets. Furthermore, for different network architectures, CNN-based networks achieve better performance than the transformer-based models (ViT). According to the analysis of Park and Kim [49] and Li et al. [39], this may be because multi-head attentions (MHA) are low-pass filters with a shape bias, while convolutions are high-pass filters with a texture bias. As a result, ViT more likely focuses on the invariant shapes between CIFAR10 and the corrupted-CIFAR10 to make the two datasets less discriminative.

## 5.4 Case Study: Learning with Noisy Labels

R-Div is a useful tool to quantify the discrepancy between two probability distributions. We demonstrate how it can learn robust networks on data with noisy labels by distinguishing samples with noisy labels from clean samples in a batch. Furthermore, we explain how the discrepancy between clean and noisy samples affects the classification generalization ability [58].

### 5.4.1 Setup

We adopt the benchmark dataset CIFAR10 [36]. As CIFAR10 contains clean samples, following the setup [51], we corrupt its training samples manually, i.e., flipping partial clean labels to noisy labels. Specifically, we consider symmetry flipping [56] and pair flipping [21]. For corrupting a label, symmetry flipping transfers it to another random label, and pair flipping transfers it to a specific label. Following the setup [29], we adopt 0.2 noise rate, i.e., the labels of 20% samples will be flipped. The classification accuracy is evaluated on clean test samples.

We apply R-Div to train a robust DNN on a corrupted dataset. The training strategy separates clean and noisy samples by maximizing the discrepancy estimated by R-Div. The ground truth labels

Table 5: The average test power on a dataset and its corrupted variants at the significant level $\alpha = 0.05$.

| Dataset | Corruption | 0.5% | 1% | 2% | 5% |
|---------|-----------|------|----|----|-----|
| | | | H-Div/ R-Div | | |
| CIFAR10 ($N = 1000$) | Gauss | 0.000 / 0.000 | 0.195 / **0.219** | 0.368 / **0.411** | 0.710 / **0.748** |
| | Snow | 0.000 / 0.000 | 0.096 / **0.189** | 0.251 / **0.277** | 0.366 / **0.416** |
| | Speckle | 0.000 / 0.000 | 0.275 / **0.340** | 0.458 / **0.513** | 0.685 / **0.771** |
| | Impulse | 0.000 / 0.000 | 0.127 / **0.185** | 0.277 / **0.326** | 0.629 / **0.652** |
| ImageNet ($N = 5000$) | Gauss | 0.000 / 0.000 | 0.680 / **0.722** | **1.000 / 1.000** | **1.000 / 1.000** |
| | Snow | 0.000 / 0.000 | 0.774 / **0.816** | 0.785 / **0.836** | **1.000 / 1.000** |
| | Speckle | 0.000 / 0.000 | 0.586 / **0.624** | 0.762 / **0.804** | **1.000 / 1.000** |
| | Impulse | 0.000 / 0.000 | 0.339 / **0.485** | 0.545 / **0.635** | **1.000 / 1.000** |

Table 6: The average test power on advanced network architectures.

| N | ResNet18 | SENet | ConvNeXt | ViT |
|---|----------|-------|----------|-----|
| | | H-Div / R-Div | | |
| 1000 | 0.116 / **0.252** | 0.201 / **0.296** | 0.186 / **0.322** | 0.119 / **0.211** |
| 5000 | 0.622 / **0.748** | 0.478 / **0.790** | 0.778 / **0.844** | 0.492 / **0.651** |
| 10000 | **1.000 / 1.000** | **1.000 / 1.000** | **1.000 / 1.000** | **1.000 / 1.000** |

for noisy samples are unreliable. Therefore, these labels can be treated as complementary labels representing the classes that the samples do not belong to. Accordingly, the training strategy optimizes the losses of ground truth and complementary labels for clean and noisy samples, respectively.

We assume that clean and noisy samples satisfy different distributions, i.e., clean and noisy distributions. Specifically, we apply R-Div to detect clean and noisy samples from a batch. Accordingly, we separate the batch samples $\mathcal{B}$ into predicted clean samples $\mathcal{B}_c$ and predicted noisy samples $\mathcal{B}_n$ by maximizing R-Div. The batch and the whole dataset contain IID samples from an unknown mixture distribution about clean and noisy distributions. Both the minimum hypotheses on the batch and the whole dataset can be treated to approximate the optimal hypothesis of the mixture distribution. To prevent learning a minimum hypothesis for each given batch, we pretrain a network on the whole corrupted dataset rather than a batch and apply this pretrained network (minimum hypothesis) to separate clean and noisy samples in a batch.

Recall that R-Div calculates the empirical risk difference between two datasets under the same minimum hypothesis of their mixed dataset. Hence, to maximize the discrepancy of R-Div, we calculate the empirical risk of each sample in the batch by the pretrained network, sort the batch samples according to the empirical risks, and treat the samples with low and high empirical risks as predicted clean and noisy samples, respectively. Specifically, we separate noisy and clean samples according to the estimated noise rate $\gamma \in [0, 1]$. For clean samples with trusted ground truth labels, we optimize a network by minimizing the traditional cross-entropy loss. In contrast, the labels for noisy samples are untrusted. However, their noisy labels indicate the classes they do not belong to. Accordingly, we can treat their untrusted ground truth labels as complementary labels and optimize a network by minimizing the negative traditional cross-entropy loss on noisy samples. Thus, we retrain a robust network by optimizing the objective function $\min_{h \in \mathcal{H}} \widehat{\epsilon}_{\mathcal{B}_c}(h) - \widehat{\epsilon}_{\mathcal{B}_n}(h)$ for each batch, where the loss function $l$ is the traditional cross-entropy loss. In this work, we adopt ResNet18 [22] as the backbone. In the pretraining and retraining processes, the learning rate [60] starts at 0.1 and is divided by 10 after 100 and 150 epochs. The batch size is 128, and the number of epochs is 200. The hyper-parameter $\gamma$ varies from 0.1 to 0.7 with a 0.1 step size.

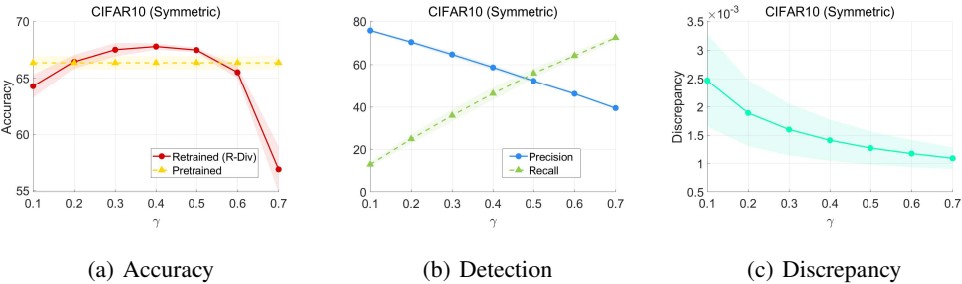

|  (a) Accuracy | (b) Detection | (c) Discrepancy |

Figure 2: Results on CIFAR10 with symmetry flipping. All values are averaged over five trials. **Left:** Classification accuracy of pretrained and retrained networks. **Middle:** Precision and recall rates of detecting clean and noisy samples. **Right:** Discrepancy between predicted clean and noisy samples.

### 5.4.2 Experimental Results

We evaluate the classification accuracy on clean test samples, the precision and recall rates of detecting clean and noisy samples, and the discrepancy between predicted clean and noisy samples. The results are reported in Figure 2 and Figure 6 (see Appendix D.3). Recall that the noise rate is 0.2. Figure 2(a) and Figure 6(a) show that the network retrained with R-Div improves the classification performance over the pretrained network when the estimated noise rate $\gamma \in [0.2, 0.5]$. The retrained network achieves the best performance when the estimated noise rate is nearly twice as much as the real noise rate. To the best of our knowledge, how to best estimate the noise rate remains an open problem and severely limits the practical application of the existing algorithms. However, our method merely requires an approximate estimation which can be twice as much as the real noise rate.

Further, Figure 2(b) and Figure 6(b) show that there is a point of intersection between the precision and recall curves at $\gamma = 0.5$. The aforementioned results indicate that $\gamma \leq 0.5$ can lead to improved performance. Accordingly, we know that the detection precision is more essential than its recall rate. Specifically, a larger estimated noise rate ($\gamma \geq 0.5$) indicates that more clean samples are incorrectly recognized as noisy samples. In contrast, an extremely lower estimated noise rate which is lower than the real one indicates that more noisy samples are incorrectly recognized as clean samples. Both situations can mislead the network in learning to classify clean samples.

In addition, Figure 2(c) and Figure 6(c) show that a larger $\gamma$ leads to a smaller discrepancy. Because the majority of training samples are clean, most of the clean samples have small test losses. A larger $\gamma$ causes more clean samples to be incorrectly recognized as noisy samples, which decreases the test loss of the set of predicted noisy samples. Thus, the empirical estimator will decrease. Based on the results reported in Figure 2(a) and Figure 6(a), the largest discrepancy cannot lead to improved classification performance because most of the noisy samples are incorrectly recognized as clean samples. In addition, we observe that the retrained network obtains improved classification performance when the estimated discrepancy approximates the mean value $1.5 \times 10^{-3}$, which can be used as a standard to select an estimated noise rate $\gamma$ for retraining a robust network.

## 6  Conclusions

We tackle the crucial question of whether two probability distributions are identically distributed for a specific learning model using a novel and effective metric, R-divergence. Unlike existing divergence measures, R-divergence identifies a minimum hypothesis from a dataset that combines the two given datasets, and then uses this to compute the empirical risk difference between the individual datasets. Theoretical analysis confirms that R-divergence, serving as an empirical estimator, uniformly converges and is bounded by the discrepancies between the two distributions. Empirical evaluations reveal that R-divergence outperforms existing methods across both supervised and unsupervised learning paradigms. Its efficacy in handling learning with noisy labels further underscores its utility in training robust networks. If the samples from the two datasets are deemed identically distributed, a subsequent question to be investigated is whether these samples are also dependent.

## Acknowledgments and Disclosure of Funding

This work was supported in part by the Australian Research Council Discovery under Grant D-P190101079 and in part by the Future Fellowship under Grant FT190100734.

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

# A Proofs

Section A.1 presents the lemmas used to prove the main results. Section A.2 presents the main results and their detailed proofs.

## A.1 Preliminaries

**Lemma A.1** ([53], Theorem 26.5). *With probability of at least $1 - \delta$, for all $h \in \mathcal{H}$ ,*

$$|\epsilon_u(h) - \widehat{\epsilon}_{\widehat{u}}(h)| \leq 2\mathcal{R}_H^l(\widehat{u}) + 4c\sqrt{\frac{\ln(4/\delta)}{N}}.$$

**Lemma A.2** ([53], Theorem 26.5). *With probability of at least $1 - \delta$, for all $h \in \mathcal{H}$,*

$$\epsilon_u(\widehat{h}_u) - \epsilon_u(h_u^*) \leq 2\mathcal{R}_H^l(\widehat{u}) + 5c\sqrt{\frac{\ln(8/\delta)}{N}}.$$

**Lemma A.3** ([44], Talagrand's contraction lemma). *For any $L$-Lipschitz loss function $l(\cdot, \cdot)$, we obtain,*

$$\mathcal{R}(l \circ \mathcal{H} \circ \widehat{u}) \leq L\mathcal{R}(\mathcal{H} \circ \widehat{u}),$$

*where $\mathcal{R}_{\mathcal{H}}^l(\widehat{u}) = \frac{1}{2N}\mathbb{E}_{\boldsymbol{\sigma} \sim \{\pm 1\}^{2N}}\left[\sup_{h \in \mathcal{H}} \sum_{x \in \widehat{u}} \sigma_i h(x_i)\right].$*

**Lemma A.4** ([18], Theorem 1). *Let $\mathcal{H}$ be the class of real-valued networks of depth $D$ over the domain $\mathcal{X}$. Assume the Frobenius norm of the weight matrices are at most $M_1, \ldots, M_d$. Let the activation function be 1-Lipschitz, positive-homogeneous, and applied element-wise (such as the ReLU). Then,*

$$\mathbb{E}_{\sigma \in \{\pm 1\}^N}\left[\sup_{h \in \mathcal{H}} \sum_{i=1}^{N} \sigma_i h(x_i)\right] \leq \sqrt{N}B(\sqrt{2D \ln 2} + 1)\prod_{i=1}^{D} M_i.$$

## A.2 Main Results

**Theorem 4.1.** *For any $x \in \mathcal{X}$, $h \in \mathcal{H}$ and $a \in \mathcal{A}$, assume we have $|l(h(x), a(x))| \leq c$. Then, with probability at least $1 - \delta$, we can derive the following general bound,*

$$|\mathcal{D}_R(p\|q) - \widehat{\mathcal{D}}_R(\widehat{p}\|\widehat{q})| \leq 2\mathcal{D}_{\mathrm{L}}(p\|q) + 2\mathcal{R}_{\mathcal{H}}^l(\widehat{p}) + 2\mathcal{R}_{\mathcal{H}}^l(\widehat{q}) + 12c\sqrt{\frac{\ln(8/\delta)}{N}}.$$

*Proof.* We observe that

$$
\begin{aligned}
&|\mathcal{D}_{\mathrm{R}}(p\|q) - \widehat{\mathcal{D}}_{\mathrm{R}}(\widehat{p}\|\widehat{q})| \\
&= \left||\epsilon_q(h_u^*) - \epsilon_p(h_u^*)| - |\widehat{\epsilon}_{\widehat{q}}(\widehat{h}_u) - \widehat{\epsilon}_{\widehat{p}}(\widehat{h}_u)|\right| \\
&\leq \left|\epsilon_q(h_u^*) - \epsilon_p(h_u^*,) - \widehat{\epsilon}_{\widehat{q}}(\widehat{h}_u) + \widehat{\epsilon}_{\widehat{p}}(\widehat{h}_u)\right| \\
&= \left|\epsilon_q(h_u^*) - \epsilon_p(h_u^*) + \widehat{\epsilon}_{\widehat{p}}(\widehat{h}_u) - \epsilon_p(\widehat{h}_u) + \epsilon_p(\widehat{h}_u) - \epsilon_q(\widehat{h}_u) + \epsilon_q(\widehat{h}_u) - \widehat{\epsilon}_{\widehat{q}}(\widehat{h}_u)\right| \\
&\leq |\epsilon_q(h_u^*) - \epsilon_p(h_u^*)| + \left|\epsilon_p(\widehat{h}_u) - \epsilon_q(\widehat{h}_u)\right| + \left|\widehat{\epsilon}_{\widehat{p}}(\widehat{h}_u) - \epsilon_p(\widehat{h}_u)\right| + \left|\epsilon_q(\widehat{h}_u) - \widehat{\epsilon}_{\widehat{q}}(\widehat{h}_u)\right| \\
&\leq 2\mathcal{D}_{\mathrm{L}}(p\|q) + \left|\widehat{\epsilon}_{\widehat{p}}(\widehat{h}_u) - \epsilon_p(\widehat{h}_u)\right| + \left|\epsilon_q(\widehat{h}_u) - \widehat{\epsilon}_{\widehat{q}}(\widehat{h}_u)\right|.
\end{aligned}
\tag{7}
$$

The first two inequalities are owing to the triangle inequality, and the third inequality is due to the definition of L-divergence Eq.(5). We complete the proof by applying Lemma A.1 to bound $\left|\widehat{\epsilon}_{\widehat{p}}(\widehat{h}_u) - \epsilon_p(\widehat{h}_u)\right|$ and $\left|\epsilon_q(\widehat{h}_u) - \widehat{\epsilon}_{\widehat{q}}(\widehat{h}_u)\right|$ in Eq.(7). $\qquad\square$

**Corollary 4.2.** *Following the conditions of Theorem 4.1, the upper bound of* $\sqrt{\mathrm{Var}\left[\widehat{\mathcal{D}}_R(\widehat{p}\|\widehat{q})\right]}$ *is*

$$\frac{34c}{\sqrt{N}} + \sqrt{2\pi}\left(2\mathcal{D}_{\mathrm{L}}(p\|q) + 2\mathcal{R}_{\mathcal{H}}^l(\widehat{p}) + 2\mathcal{R}_{\mathcal{H}}^l(\widehat{q})\right).$$

*Proof.* For convenience, we assume $\mathcal{C} = 2\mathcal{D}_{\mathrm{L}}(p\|q) + 2\mathcal{R}_{\mathcal{H}}^l(\widehat{p}) + 2\mathcal{R}_{\mathcal{H}}^l(\widehat{q})$. Following to proof of Corollary 1 in [62], we have □

$$
\begin{aligned}
\mathrm{Var}\left[\widehat{\mathcal{D}}_R(\widehat{p}\|\widehat{q})\right] \leq & \mathbb{E}\left[\left(\mathcal{D}_R(p\|q) - \widehat{\mathcal{D}}_R(\widehat{p}\|\widehat{q})\right)^2\right] \\
= & \int_{t=0}^{\infty} \Pr\left[\left|\mathcal{D}_R(p\|q) - \widehat{\mathcal{D}}_R(\widehat{p}\|\widehat{q})\right| \geq \sqrt{t}\right] dt \\
\overset{s=\sqrt{t}}{=} & \int_{s=0}^{\infty} \Pr\left[\left|\mathcal{D}_R(p\|q) - \widehat{\mathcal{D}}_R(\widehat{p}\|\widehat{q})\right| \geq s\right] 2s\, ds \\
\leq & \int_{s=0}^{\mathcal{C}} 2s\, ds + \int_{s=0}^{\infty} \Pr\left[\left|\widehat{\mathcal{D}}_R(\widehat{p}\|\widehat{q}) - \mathcal{D}_R(p\|q)\right| \geq \mathcal{C} + s\right] 2(\mathcal{C}+s) ds \\
\leq & \mathcal{C}^2 + 16 \int_{s=0}^{\infty} (\mathcal{C}+s) e^{\frac{-Ns^2}{144c^2}} ds \qquad\qquad (8) \\
= & \mathcal{C}^2 + 16\mathcal{C} \int_{s=0}^{\infty} e^{\frac{-Ns^2}{144c^2}} ds + 16 \int_{s=0}^{\infty} s e^{\frac{-Ns^2}{144c^2}} ds \\
\overset{t=\frac{s}{c}\sqrt{\frac{N}{144}}}{=} & \mathcal{C}^2 + 16\mathcal{C}c\sqrt{\frac{a}{N}} \int_{t=0}^{\infty} e^{-t^2} dt + \frac{2304c^2}{N} \int_{s=0}^{\infty} t e^{-t^2} dt \\
= & \mathcal{C}^2 + 96\mathcal{C}c\sqrt{\frac{\pi}{N}} + \frac{1152c^2}{N} \\
\leq & \left(\frac{34c}{\sqrt{N}} + \mathcal{C}\sqrt{2\pi}\right)^2,
\end{aligned}
$$

where the third inequality is due to

$$\Pr\left[\left|\mathcal{D}_R(p\|q) - \widehat{\mathcal{D}}_R(\widehat{p}\|\widehat{q})\right| \geq \mathcal{C} + s\right] \leq 8e^{\frac{-Ns^2}{144c^2}}, \qquad\qquad (9)$$

obtained from the results of Theorem 4.1.

**Proposition 4.3.** *Based on the conditions of Theorem 4.1, we assume $\mathcal{H}$ is the class of real-valued networks of depth $D$ over the domain $\mathcal{X}$. Let the Frobenius norm of the weight matrices be at most $M_1, \ldots, M_D$, the activation function be 1-Lipschitz, positive-homogeneous and applied element-wise (such as the ReLU). Then, with probability at least $1 - \delta$, we have,*

$$|\mathcal{D}_R(p\|q) - \widehat{\mathcal{D}}_R(\widehat{p}\|\widehat{q})| \leq 2\mathcal{D}_{\mathrm{L}}(p\|q) + \frac{4LB(\sqrt{2D\ln 2}+1)\prod_{i=1}^{D} M_i}{\sqrt{N}} + 12c\sqrt{\frac{\ln(8/\delta)}{N}}.$$

*Proof.* We complete the proof by applying Lemma A.3 and Lemma A.4 to bound the Rademacher complexity of deep neural networks in Theorem 4.1. □

**Corollary 4.4.** *Following the conditions of Proposition 4.3, with probability at least $1 - \delta$, we have the following conditional bounds if $\mathcal{D}_{\mathrm{L}}(p\|q) \leq \left|\epsilon_u(h_u^*) - \epsilon_u(\widehat{h}_u)\right|$,*

$$|\mathcal{D}_R(p\|q) - \widehat{\mathcal{D}}_R(\widehat{p}\|\widehat{q})| \leq \frac{8LB(\sqrt{2D\ln 2}+1)\prod_{i=1}^{D} M_i}{\sqrt{N}} + 22c\sqrt{\frac{\ln(16/\delta)}{N}}.$$

*Proof.* Following the proof of Theorem 4.1, we have

$$
\begin{aligned}
&|\mathcal{D}_{\mathrm{R}}(p\|q) - \widehat{\mathcal{D}}_{\mathrm{R}}(\widehat{p}\|\widehat{q})| \\
&\leq \left|\epsilon_q(h_u^*) - \epsilon_p(h_u^*) + \widehat{\epsilon}_{\widehat{p}}(\widehat{h}_u) - \epsilon_p(\widehat{h}_u)\right| + \left|\widehat{\epsilon}_{\widehat{p}}(\widehat{h}_u) - \epsilon_p(\widehat{h}_u)\right| + \left|\epsilon_q(\widehat{h}_u) - \widehat{\epsilon}_{\widehat{q}}(\widehat{h}_u)\right| \\
&= \max\left(\left|\epsilon_q(h^*) - \epsilon_p(h^*) + \epsilon_p(\widehat{h}) - \epsilon_q(\widehat{h})\right|, \left|(\epsilon_q(h^*) + \epsilon_p(h^*)) - \left(\epsilon_p(\widehat{h}) - \epsilon_q(\widehat{h})\right)\right|\right) \\
&\quad + \left|\widehat{\epsilon}_{\widehat{p}}(\widehat{h}_u) - \epsilon_p(\widehat{h}_u)\right| + \left|\epsilon_q(\widehat{h}_u) - \widehat{\epsilon}_{\widehat{q}}(\widehat{h}_u)\right| \\
&\leq 2\max\left(D_{\mathrm{L}}(p\|q), \left|\epsilon_u(h^*) - \epsilon_u(\widehat{h})\right|\right) + \left|\widehat{\epsilon}_{\widehat{p}}(\widehat{h}_u) - \epsilon_p(\widehat{h}_u)\right| + \left|\epsilon_q(\widehat{h}_u) - \widehat{\epsilon}_{\widehat{q}}(\widehat{h}_u)\right| \\
&\leq 2\left|\epsilon_u(h^*) - \epsilon_u(\widehat{h})\right| + \left|\widehat{\epsilon}_{\widehat{p}}(\widehat{h}_u) - \epsilon_p(\widehat{h}_u)\right| + \left|\epsilon_q(\widehat{h}_u) - \widehat{\epsilon}_{\widehat{q}}(\widehat{h}_u)\right|.
\end{aligned}
\tag{10}
$$

The first two inequalities are owing to the triangle inequality, and the third inequality is due to the given condition $\mathcal{D}_{\mathrm{L}}(p\|q) \leq \left|\epsilon_u(h_u^*) - \epsilon_u(\widehat{h}_u)\right|$. We complete the proof by applying Lemma A.2 to bound $\left|\epsilon_u(h^*) - \epsilon_u(\widehat{h})\right|$ and Lemma A.1 to bound $\left|\widehat{\epsilon}_{\widehat{p}}(\widehat{h}_u) - \epsilon_p(\widehat{h}_u)\right|$ and $\left|\epsilon_q(\widehat{h}_u) - \widehat{\epsilon}_{\widehat{q}}(\widehat{h}_u)\right|$ in Eq.(7). $\qquad\square$

**Corollary 4.5.** *Following the conditions of Proposition 4.3, as $N \to \infty$, we have,*

$$
\widehat{\mathcal{D}}_{\mathrm{R}}(\widehat{p}\|\widehat{q}) \leq 2\mathcal{D}_{\mathrm{L}}(p\|q) + \mathcal{D}_{\mathrm{R}}(p\|q).
$$

*Proof.* Based on the result on Proposition 4.3, for any $\delta \in (0,1)$, we know that

$$
\frac{4LB(\sqrt{2D\ln 2}+1)\prod_{i=1}^{D} M_i}{\sqrt{N}} + 12c\sqrt{\frac{\ln(8/\delta)}{N}} \to 0,
\tag{11}
$$

when $N \to \infty$. We complete the proof by applying the triangle inequality. $\qquad\square$

# B  Algorithm Procedure

## B.1  Training Procedure of R-Div

---
**Algorithm 1** Estimate model-oriented distribution discrepancy by R-divergence

---
**Input:** two datasets $\widehat{p}$ and $\widehat{q}$ and a learning model $\mathcal{T}$ with hypothesis space $\mathcal{H}$ and loss function $l$
Generate the merged dataset: $\widehat{u} = \widehat{p} \cup \widehat{q}$
Learn the minimum hypothesis on the mixed data:

$$
\widehat{h}_u \in \arg\min_{h \in \mathcal{H}} \widehat{\epsilon}_{\widehat{u}}(h)
$$

Evaluate the empirical risks: $\widehat{\epsilon}_{\widehat{p}}(\widehat{h}_{\widehat{u}})$ and $\widehat{\epsilon}_{\widehat{q}}(\widehat{h}_{\widehat{u}})$
Estimate the R-divergence as the discrepancy:

$$
\widehat{\mathcal{D}}_{\mathrm{R}}(\widehat{p}\|\widehat{q}) = |\widehat{\epsilon}_{\widehat{p}}(\widehat{h}_{\widehat{u}}) - \widehat{\epsilon}_{\widehat{q}}(\widehat{h}_{\widehat{u}})|
$$

**Output:** empirical estimator $\widehat{\mathcal{D}}_{\mathrm{R}}(\widehat{p}\|\widehat{q})$

---

## B.2  Calculation Procedure of the Average Test Power

---

**Algorithm 2** Calculate the average test power of R-divergence

---

**Input:** datasets $\widehat{p}$ and $\widehat{q}$, empirical estimator $\widehat{\mathcal{D}}_{\mathrm{R}}(\cdot\|\cdot)$, $\alpha$, $K$, $Z$
**for** $k = 1$ **to** $K$ **do**
    **for** $z = 1$ **to** $Z$ **do**
        **if** $z == 1$ **then**
            $(\widehat{p}^1, \widehat{q}^1) = (\widehat{p}, \widehat{q})$
        **else**
            Generate $(\widehat{p}^z, \widehat{q}^z)$ by uniformly randomly swapping samples between $\widehat{p}$ and $\widehat{q}$
        **end if**
        Calculate $\widehat{\mathcal{D}}_{\mathrm{R}}(\widehat{p}^z\|\widehat{q}^z)$
    **end for**
    Obtain $\mathcal{G} = \{\widehat{\mathcal{D}}_{\mathrm{R}}(\widehat{p}^z\|\widehat{q}^z)\}_{z=1}^Z$
    $r_k = \left(\widehat{\mathcal{D}}_{\mathrm{R}}(\widehat{p}\|\widehat{q}) \text{ is in the top } \alpha\text{-quantile among } \mathcal{G}\right)$
**end for**
**Output:** average test power $\frac{\sum_{k=1}^K r_k}{K}$

---

Table 7: Method Comparison.

| Methods | Discrepancy | Remarks |
|---|---|---|
| ME | $\sqrt{\frac{1}{J}\sum_{j=1}^J \left(\mu_p(T_j) - \mu_q(T_j)\right)^2}$ | I:$\mu_p(t) = \int k(x,t)dp(x)$
II:$k(x,t) = \exp(-\|x-y\|^2/\gamma^2)$
III:$T$ is drawn from an absolutely continuous distribution. |
| SCF | $\sqrt{\frac{1}{2J}\sum_{j=1}^2 \left(z_p^{\sin}(T_j) - z_p^{\sin}(T_j)\right)^2 + \left(z_p^{\cos}(T_j) - z_p^{\cos}(T_j)\right)^2}$ | I:$z_p^{\sin}(t) = \int \kappa(x)\sin(x^T t)dp(x)$
II:$z_p^{\cos}(t) = \int \kappa(x)\cos(x^T t)dp(x)$
III:$\kappa(x)$ is the Fourier transform of a kernel. |
| C2STS-S | $\frac{1}{|\hat{p}_{\mathrm{te}}|}\sum_{x\in\hat{p}_{\mathrm{te}}} f_w(x) - \frac{1}{|\hat{q}_{\mathrm{te}}|}\sum_{x'\in\hat{q}_{\mathrm{te}}} f_w(x')$ | $I : \hat{p}_{\mathrm{tr}}, \hat{p}_{\mathrm{te}} \sim p, \hat{q}_{\mathrm{tr}}, \hat{q}_{\mathrm{te}} \sim q$
II: $f_w$ is a binary classifier to separate $\hat{p}_{\mathrm{tr}}$ and $\hat{q}_{\mathrm{tr}}$
III: Samples from $p$ and $q$ are labeled with 0 and 1, respectively. |
| C2ST-L | $\frac{1}{|\hat{p}_{\mathrm{te}}|+|\hat{q}_{\mathrm{te}}|}\sum_{x\in\hat{p}_{\mathrm{te}}} g_w(x,0) + \sum_{x'\in\hat{q}_{\mathrm{te}}} g_w(x',1)$ | IV: $g_w(x,y) = \mathbb{I}\left[\mathbb{I}\left(f_w(x) > \frac{1}{2}\right) = y\right]$ |
| MMD-O | $\sqrt{\mathbb{E}\left[k(x,x') + k(y,y') - 2k(x,y)\right]},$
$x, x \sim p, y, y' \sim q$ | I:$k(x,y)$ is a simple kernal. |
| MMD-D | $\sqrt{\mathbb{E}\left[k_w(x,x') + k_w(y,y') - 2k_w(x,y)\right]},$
$x, x \sim p, y, y' \sim q$ | I:$k_w(x,y) = [(1-\epsilon)k_1(\phi_w(x), \phi_w(y)) + \epsilon]k_2(x,y)$
II:$\phi_w$ is deep network with parameters $w$
III:$k_1(x,y) = \exp(-\|x-y\|^2/\gamma_1^2)$
IV:$k_2(x,y) = \exp(-\|x-y\|^2/\gamma_2^2)$ |
| H-Div | $\phi(\epsilon_u(h_u^*) - \epsilon_p(h_p^*), \epsilon_u(h_u^*) - \epsilon_q(h_q^*))$ | I:$h_u^* \in \arg\min_{h\in\mathcal{H}} \epsilon_u(h)$
II:$h_q^* \in \arg\min_{h\in\mathcal{H}} \epsilon_q(h)$
III:$\epsilon_u(h) = \mathbb{E}_{x\sim u}l(h(x), a(x))$ |
| R-Div | $|\epsilon_p(h_u^*) - \epsilon_q(h_u^*)|$ | IV:$\epsilon_q(h) = \mathbb{E}_{x\sim q}l(h(x), a(x))$
V:$\phi(\theta, \lambda) = \frac{\theta+\lambda}{2}$ or $\max(\theta, \lambda)$ |

## C  Compared Methods

Mean embedding (ME) [11] and smooth characteristic functions (SCF) [30] are the state-of-the-art methods using differences in Gaussian mean embeddings at a set of optimized points and frequencies, respectively. Classifier two-sample tests, including C2STS-S [43] and C2ST-L [10], apply the classification accuracy of a binary classifier to distinguish between the two distributions. The binary classifier treats samples from one dataset as positive and the other dataset as negative. These two methods assume that the binary classifier cannot distinguish these two kinds of samples if their distributions are identical. Differently, C2STS-S and C2ST-L apply the test error and the test error gap to evaluate the discrepancy, respectively. MMD-O [20] measures the maximum mean discrepancy (MMD) with a Gaussian Kernel [19], and MMD-D [41] improves the performance of MMD-O by replacing the Gaussian Kernel with a learnable deep kernel. H-Divergence (H-Div) [62] learns optimal hypotheses for the mixture distribution and each individual distribution for the specific model, assuming that the expected risk of training data on the mixture distribution is higher than that on each individual distribution if the two distributions are identical. The equations of these compared methods are presented in Figure 7.

# D   Additional Experimental Results

## D.1   Benchmark Dataset

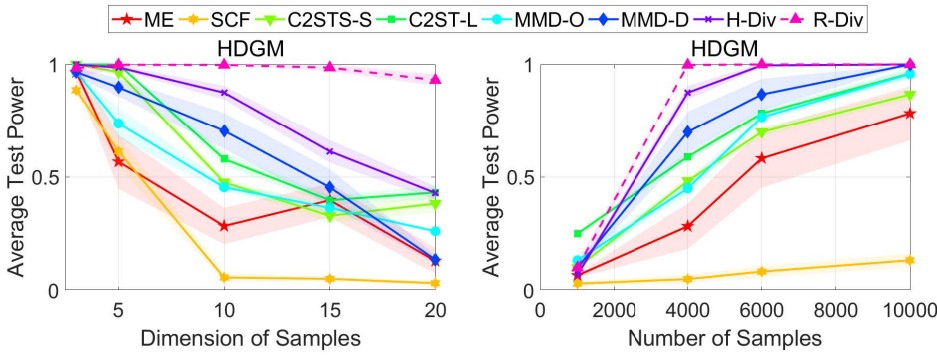

Figure 3: The average test power on HDGM with the significant level $\alpha = 0.05$. Left panel: results with the same sample size (4,000) and different feature dimensions. Right panel: results with the same feature dimensions (10) and different sample sizes.

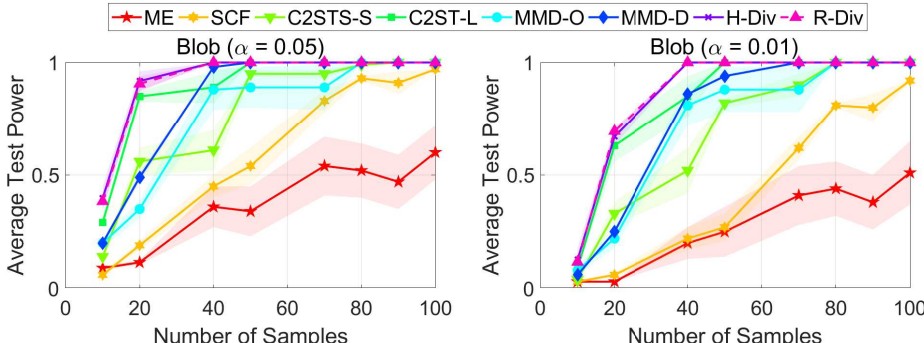

Figure 4: The average test power on Blob at the significant levels $\alpha = 0.05$ and $\alpha = 0.01$.

Table 8: The average test power $\pm$ standard error with the significant level $\alpha = 0.05$ on MNIST. $N$ represents the number of samples in each given dataset, and boldface values represent the relatively better discrepancy estimation.

| N | 200 | 400 | 600 | 800 | 1000 | Avg. |
|---|---|---|---|---|---|---|
| ME | $0.414\pm_{0.050}$ | $0.921\pm_{0.032}$ | $\mathbf{1.000}\pm_{\mathbf{0.000}}$ | $\mathbf{1.000}\pm_{\mathbf{0.000}}$ | $\mathbf{1.000}\pm_{\mathbf{0.000}}$ | 0.867 |
| SCF | $0.107\pm_{0.018}$ | $0.152\pm_{0.021}$ | $0.294\pm_{0.008}$ | $0.317\pm_{0.017}$ | $0.346\pm_{0.019}$ | 0.243 |
| C2STS-S | $0.193\pm_{0.037}$ | $0.646\pm_{0.039}$ | $\mathbf{1.000}\pm_{\mathbf{0.000}}$ | $\mathbf{1.000}\pm_{\mathbf{0.000}}$ | $\mathbf{1.000}\pm_{\mathbf{0.000}}$ | 0.768 |
| C2ST-L | $0.234\pm_{0.031}$ | $0.706\pm_{0.047}$ | $0.977\pm_{0.012}$ | $\mathbf{1.000}\pm_{\mathbf{0.000}}$ | $\mathbf{1.000}\pm_{\mathbf{0.000}}$ | 0.783 |
| MMD-O | $0.188\pm_{0.010}$ | $0.363\pm_{0.017}$ | $0.619\pm_{0.021}$ | $0.797\pm_{0.015}$ | $0.894\pm_{0.016}$ | 0.572 |
| MMD-D | $0.555\pm_{0.044}$ | $0.996\pm_{0.004}$ | $\mathbf{1.000}\pm_{\mathbf{0.000}}$ | $\mathbf{1.000}\pm_{\mathbf{0.000}}$ | $\mathbf{1.000}\pm_{\mathbf{0.000}}$ | 0.910 |
| H-Div | $\mathbf{1.000}\pm_{\mathbf{0.000}}$ | $\mathbf{1.000}\pm_{\mathbf{0.000}}$ | $\mathbf{1.000}\pm_{\mathbf{0.000}}$ | $\mathbf{1.000}\pm_{\mathbf{0.000}}$ | $\mathbf{1.000}\pm_{\mathbf{0.000}}$ | $\mathbf{1.000}$ |
| R-Div | $\mathbf{1.000}\pm_{\mathbf{0.000}}$ | $\mathbf{1.000}\pm_{\mathbf{0.000}}$ | $\mathbf{1.000}\pm_{\mathbf{0.000}}$ | $\mathbf{1.000}\pm_{\mathbf{0.000}}$ | $\mathbf{1.000}\pm_{\mathbf{0.000}}$ | $\mathbf{1.000}$ |

## D.2 PACS Dataset

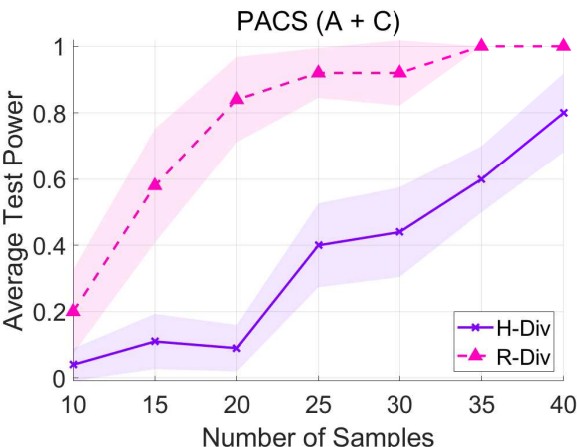

Figure 5: The average test power at the significant level $\alpha = 0.05$ on the art painting and cartoon domains of PACS.

## D.3 Learning with Noisy Labels

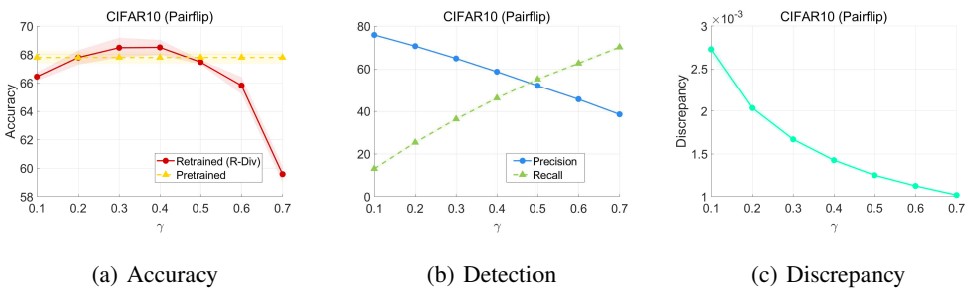

(a) Accuracy           (b) Detection           (c) Discrepancy

Figure 6: Results on CIFAR10 with pair flipping. All values are averaged over five trials. **Left:** Classification accuracy of pretrained and retrained networks. **Middle:** Precision and recall rates of detecting clean and noisy samples. **Right:** Discrepancy between predicted clean and noisy samples.

