# OpenReview forum: "R-divergence for Estimating Model-oriented Distribution Discrepancy"
_NeurIPS.cc/2023/Conference — NeurIPS 2023 poster_

### Official Review · Reviewer_Z32S · 2023-07-04

**Soundness:** 4 excellent
**Presentation:** 3 good
**Contribution:** 3 good
**Rating:** 7
**Confidence:** 4

**Summary:**

A new divergence metric, called R-divergence, is proposed in the paper. This metric states that two distributions are identical if the optimal hypothesis on the mixture distribution has the same expected risk on each individual distribution.
If two distributions are the same, then the optimal hypothesis on the mixture distribution is the same as the one for the individual distributions. This means that the expected risk is similar for each individual distribution. However, if the distributions are different, then the optimal hypothesis on the mixture distribution would behave differently for the individual distributions. This would result in different expected risks for the individual distributions.
The main difference between the proposed R-divergence and the existing H-divergence is that R-divergence calculates the minimum hypothesis and the empirical risk on different datasets. This prevents R-divergence from the overfitting issue that H-divergence suffers from.

The paper also utilizes R-divergence to learn with noisy labels, whereby R-divergence is utilized per batch to estimate the clean and noisy samples, and then the network is retrained with minimizing the cross entropy over the clean batch while maximizing it over the corrupted samples.

The paper conducts two-sample tests over different datasets and architecture and achieves superior numbers than H-divergence.

**Strengths:**

1. The paper is very well written and is easy to follow. The difference with H-divergence is well highlighted too in the paper.

2. The paper achieves strong results over the baseline divergence metrics over different datasets and with different model architectures.

3. I find the comparison between ViT and CNN architectures particularly interesting, whereby ViT architectures achieve worse test power than CNN ones, indicating that they are learning shape-biased features which makes it hard to discriminate between CIFAR-10 and corrupted CIFAR-10.

4. Using R-divergence to improve the performance of models under corrupted labels is a nice example of how these divergence metrics could be useful.

**Weaknesses:**

1. The comparison with CIFAR-10 and corrupted CIFAR-10 is conducted with only one type of corruption when H-divergence considers a variety of corruptions. Can the authors do more detailed experimentation with different corruptions as done in H-divergence.

2. Can the authors also try to do some experiments with ImageNet and its corrupted variants to more justify their proposed method. Since the experiment needs to be done with a limited number of samples, this experiments shouldn't be too computationally difficult.


**Questions:**

I have already mentioned the questions in the weakness section.

---

> ### Author Rebuttal · Authors · 2023-08-05
>
> We are grateful for your comments and address them below.
>
> **Q1. Different corruptions:**
>
> Thank you for your insightful comments and suggestions. Following the construction methods of corrupted samples in [56] and the setup of H-divergence [63], we have performed experiments of R-divergence with partial corruption types (Gauss, Snow, Speckle, and Impulse) at four corruption levels (0.5%, 1%, 2%, and 5%) on CIFAR10 and adopted Resnet18 as the backbone. Evaluating the discrepancies for data with subtle corruptions is challenging because the corresponding probability distributions are not significantly different. That is why we apply a large scale (N = 5000) for CIFAR10 vs. Corrupted-CIFAR10 and R-divergence achieves 0 test power when the corruption level is mild (e.g. 0.5%). The experimental results on different corruption types are consistent, i.e., R-divergence can obtain better performance for larger corruption levels and outperform H-divergence for all corruption types and levels.
>
> The average test power of H-divergence / R-divergence:
> | Corruption | 0.5%  | 1%    | 2%    | 5%    |
> |------------|-------|-------|-------|-------|
> | Gauss     | 0.000 / 0.000 | 0.195 / **0.219** | 0.368 / **0.411** | 0.710 / **0.748** |
> | Snow      | 0.000 / 0.000 | 0.096 / **0.189** | 0.251 / **0.277** | 0.366 / **0.416** |
> | Speckle    | 0.000 / 0.000 | 0.275 / **0.340** | 0.458 / **0.513** | 0.685 / **0.771** |
> | Impulse   | 0.000 / 0.000 | 0.127 / **0.185** | 0.277 / **0.326** | 0.629 / **0.652** |
>
> [56] Sun et al, Test-time training with self-supervision for generalization under distribution shifts, ICML, 2020.
> [63] Zhao et al, Comparing distributions by measuring differences that affect decision making, ICLR, 2022.
>
> **Q2. ImageNet and its corrupted variants:**
>
> Following the setup of the experiments on CIFAR10 vs Corrupted-CIFAR10, we have performed experiments of R-divergence on ImageNet (N = 1000) and its corrupted variants (Gauss, Snow, Speckle, and Impulse) with backbone Resnet12. The test power of H-divergence and R-divergence monotonically increases as the corruption level increases. Furthermore, R-divergence outperforms H-divergence on all the corruption types and levels. The results verify that the proposed R-divergence can be applied to estimate the distribution discrepancy for large-resolution datasets effectively.
>
> The average test power of H-divergence / R-divergence:
> | Corruption | 0.5%  | 1%    | 2%    | 5%    |
> |------------|-------|-------|-------|-------|
> | Gauss     | 0.000 / 0.000 | 0.680 / **0.722** | **1.000** / **1.000** | **1.000** / **1.000** |
> | Snow      | 0.000 / 0.000 | 0.774 / **0.816** | 0.785 / **0.836** | **1.000** / **1.000** |
> | Speckle    | 0.000 / 0.000 | 0.586 / **0.624** | 0.762 / **0.804** | **1.000** / **1.000** |
> | Impulse   | 0.000 / 0.000 | 0.339 / **0.485** | 0.545 / **0.635** | **1.000** / **1.000** |
>
> We will clarify the above in the final version.

---

> > ### Comment · Area_Chair_fkrQ · 2023-08-19
> > **Thanks for the response**
> >
> > Dear authors,
> >
> > Thanks for your efforts to clarify the concerns raised by the reviewer. Although the reviewer has not responded yet, I'd like to assure you that your response will be incorporated into the reviewers' discussion and my final recommendation.
> >
> > Best wishes,
> > AC

---

> > > ### Author Response · Authors · 2023-08-19
> > > **Thank you for your response！**
> > >
> > > We sincerely appreciate for handling the review process of our paper. We fully understand that reviewers have a busy schedule, handling numerous papers. If you or the reviewers have any new suggestions or comments on our work, please feel free to share them at any time.

---

### Official Review · Reviewer_3ypD · 2023-07-08

**Soundness:** 2 fair
**Presentation:** 3 good
**Contribution:** 3 good
**Rating:** 7
**Confidence:** 3

**Summary:**

This work proposes R-divergence as a measure of discrepancy between two distribution. The R-divergence is the difference between the empirical risks computed on two datasets using a hypothesis trained on a mixture of both datasets. The authors use the R-divergence and the permutation test to identify if two datasets are from different distributions and evaluate this method on a number of benchmarks.


**Strengths:**

The proposed R-divergence is surprisingly simple to compute: it is the difference in the empirical losses on two datasets. The method uses a trained model to define a discrepancy measure between distributions which could also include classifiers or generative models. The authors show that R-divergence can successfully separate data from different distributions and performs favorably when compared to other methods.


**Weaknesses:**

**Why is R-divergence a good measure of discrepancy?** Methods like the H-divergence [1] are rooted in principled definitions of a *probability divergence* and it satisfies properties like the triangle inequality. On the other hand R-divergence seems to be more of a heuristic and it is unclear why it is a good measure of discrepancy between two probability distributions. The authors argue that it achieves SoTA results on some benchmarks when paired with the permutation test but that does not necessarily qualify it to be a good definition of a divergence.

**Problems with the definition**: The definition seems to be reliant on the choice of the loss function and the hypothesis class. If we consider a very rich hypothesis class, we can find a hypothesis with 0 loss; this can only happen if the loss is 0 for both the distributions. Hence, the R-divergence between any two distributions will be 0 as long as we select a very rich hypothesis class. This can be problematic since the discrepancy can change based on the loss function or hypothesis class.

I worry that many results in the paper heavily depend on such choices. For example, the result on MNIST makes use of a 2-layer fully-connected network (Line 239) and I wonder if the same results will continue to hold if we consider a deep convolutional network.

**Limited insight from theory**: The main theoretical result is that the empirical R-divergence converges to the true R-divergence in the limit of infinite samples. This follows from the uniform convergence of the empirical losses to the true losses . However it is unclear why they converge quickly as claimed in the introduction (Line 73). The theory also offers no insight into why the R-divergence is better than other measures of discrepancy.

**Simple failure case of R-divergence**: Consider two distributions
$$p_1(x, y) = \frac{1}{2}\left( \delta_{(-1, -1)} + \delta_{(1, 1)} \right) \qquad \text{and} \qquad \frac{1}{2}\left( \delta_{(-100, -1)} + \delta_{(100, 1)} \right),$$
defined on $\mathbb{R} \times$ { $-1, 1$}.
The two distributions (defined on point masses) can be perfectly separated by a linear hypothesis at 0. Hence, their R-divergence is 0 if we use a linear hypothesis class, even if the two distributions are clearly different.

**Understanding when R-divergence can fail**: The R-divergence requires that the losses evaluated on the two distributions are different which is a heuristic that can only work in certain scenarios. This can occur if there is input or label noise in one of the distributions (like in Table 4, Figure 2). I worry that the authors have over-engineered to a specific set of benchmarks while ignoring many other important scenarios.

For example one could consider one distribution to be classes 0,1 from MINST and the other distribution to be classes 2,3. Can R-divergence separate the two distributions using a permutation test? Since all samples achieve a similar loss, I would expect the R-divergence to be close to 0.


[1] Zhao, Shengjia, et al. "Comparing distributions by measuring differences that affect decision making." International Conference on Learning Representations. 2021.


**Questions:**

Many of my questions have been included in a previous section but are summarized again below:

1. Why is R-divergence a good measure of discrepancy?
2. Do the results on MNIST hold if we consider a deep convolutional network?
3. Can we separate out two distributions where one contains classes 0,1 and the other contains classes 2,3 from MNIST (or CIFAR10)?
4. Do these results rely on the datasets having different scales for the losses and is this common across many datasets?



**Limitations:**

The authors can expand upon some of the limitations of their method. For example, the authors can include a discussion about how R-divergence depends on the choice of the loss function or the hypothesis class.

---

> ### Author Rebuttal · Authors · 2023-08-05
>
> We appreciate your comments, address them while also clarify potential misunderstandings below.
>
> **Q1. Advantages of R-divergence:**
>
> R-divergence is a good measure for both theoretical and practical advantages because R-divergence benefits from addressing the over-fitting issue and estimates the discrepancy with theoretical guarantees. Specifically, R-divergence learns a minimum hypothesis and calculates its empirical risk on different datasets to avoid a small empirical risk gap for significantly different distributions, as discussed in the second paragraph of Introduction. Furthermore, R-divergence can quickly converge to the sum of two terms as the sample size increases, where both terms can represent large discrepancies for two different distributions, as shown in Corollary 4.5.
>
> **Q2. Definition, Experiments with deep convolutional network, and Insight:**
>
> We believe there was a misreading of the definition. As stated in the second paragraph of Introduction, R-divergence is model-oriented for a specific learning model rather than hypothesis/loss function driven. It indicates that it is reasonable to obtain different results for different hypothesis spaces. The distributions of two given datasets can be treated as identical for some models but significantly different for others. This is because different models have different sensitivities to distribution discrepancy. For example, images of ‘Shiba Inu’ and ‘Corgi’ tend to be treated as sampled from two different distributions if we aim to learn a binary classifier to distinguish them. However, they could be treated as sampled from a mixture distribution by annotating all the images as ‘Dog’ and learn a multi-class classifier with ‘Cat’ images to learn a binary classifier to distinguish ‘Dog’ from ‘Cat’.
>
> We have performed experiments on MNIST with a deep neural network, i.e., LeNet. The results for the reconstruction task with MLP and binary cross-entropy are reported in Table 7, and the results for the classification task with LeNet and cross-entropy are reported below. The results show that the distribution discrepancies are distinct for both the reconstruction and classification tasks. The classification task achieves worst test power than the reconstruction task, which is reasonable. The classification task is less sensitive to the discrepancy because the two datasets have the same label space, and the samples of the two datasets are with covariate shifts. Furthermore, the minimum hypothesis on the mixed dataset can generalize to the samples from the two datasets, which causes the samples to tend to be drawn from a mixture distribution for this classification task.
>
> | N | 200  | 400    | 600    | 800    | 1000    | Avg.    |
> |------------|-------|-------|-------|-------|-------|-------|
> | R-Div | 0.000  | 0.210 | 0.512 | 0.976 | 1.000 | 0.539 |
>
> Proposition 4.3 shows that the empirical estimator of R-divergence can quickly converge to the discrepancy if the L-divergence is small and the sample size goes to infinity. Furthermore, Corollary 4.5 shows that R-divergence can quickly converge to the sum of L-divergence and discrepancy by increasing the sample size for a large L-divergence, where both terms can represent the difference between two distributions. Therefore, for both scenarios, R-divergence can quickly converge to a measurement representing the difference between two distributions as the sample size goes to infinity.
>
> **Q3. Failure cases and experiments on different classes:**
>
> We believe there was a misunderstanding. R-divergence can succeed in the mentioned simple case. It is impossible to find a linear hypothesis that perfectly fixes the two distributions, which indicates its empirical risks on the two datasets are not zero, and we measure the discrepancy by the empirical risk gap. A perfect hypothesis from a rich class could obtain zero empirical risks on both datasets. In this case, this hypothesis for its learning task is not sensitive to the two distributions even if the math expressions are significantly different, and this task may treat the two as one mixture distribution. For example, ‘apple’ and ‘banana’ are significantly different for a binary classification task. However, if our learning task is to learn a binary classifier to distinguish between ‘animal’ and ‘fruit’, the binary classifier can treat ‘apple’ and ‘banana’ as being drawn from the same mixture distribution and is not sensitive to their discrepancy.
>
> R-divergence can work in any scenario, as shown in Theorem 4.1, as it can adapt to any hypothesis spaces, loss functions, and target functions. Furthermore, the permutation test is applied to calculate the significant level which indicates whether the method can incorrectly decide two different distributions. The following experiment results show that R-divergence can easily separate samples with different classes from MNIST. R-divergence firstly learns a minimum hypothesis on the mixed data (classes 0~3) and calculates its empirical risks on classes 0,1 and classes 2,3, respectively. Using permutation, the minimum hypothesis is the same. However, both datasets contain samples of the four classes, which causes similar empirical risks and a small gap. Therefore, the estimated discrepancy between the original two datasets can easily be against the permutation cases. The experiments with LeNet are reported below for your reference.
>
> | N | 40  | 80    | 120    | 160    | 200    | Avg.    |
> |------------|-------|-------|-------|-------|-------|-------|
> | R-Div | 0  | 0.540 | 0.845 | 1.000 | 1.000 | 0.677  |
>
> **Q4. Data scales:**
>
> R-divergence is not sensitive to the data scales because the experiments for Imagenet vs. Its corrupted (1% Gaussian) variants with different scales lead to similar results.
>
> | Scales | 32 | 120 | 224 |
> |------------|-------|-------|-------|
> | Resize | 0.733 | 0.725 | 0.722 |
> | Crop  | 0.739 | 0.736 | 0.719 |
>
> We will clarify the above in the final version.

---

> > ### Comment · Reviewer_3ypD · 2023-08-11
> > **Thank you for the response!**
> >
> > I thank the authors for their detailed response and appreciate the time and effort. I appreciate that the authors have presented new experiments on MNIST.  I still have some questions and concerns and hope that the authors can clarify potential misunderstandings if any.
> >
> > **Q1: The name of R-divergence**
> >
> > Divergences (https://en.wikipedia.org/wiki/Divergence_(statistics)) such as KL-divergence or H-divergence, satisfy the triangle inequality and positivity. Neither of these two properties are satisfied by R-divergence which was my concern in Q1. I understand that I am being pedantic here but it would be helpful to clarify that R-divergence isn't a "divergence" from a more precise mathematical standpoint.
> >
> > edit: The abstract seems to indicate that R-divergence is a new definition for a divergence, but it seems like the main merit of the paper is to combine R-divergence with the permutation test to identify if samples are from the same/different distribution. I believe this clarification would be helpful.
> >
> > **Q3: R-divergence can work in any scenario, as shown in Theorem 4.1, as it can adapt to any hypothesis spaces, loss functions, and target functions.**
> >
> > Could the authors clarify by what they mean by "work"?  Theorem 4.1 proves that the empirical estimator converges in the asymptotic limit. But I don't understand why the value of the R-divergence is useful (to distinguish distributions or learn with noisy labels) in all scenarios.
> >
> > **Q2: R-divergence is model-oriented for a specific learning model rather than hypothesis/loss function driven. It indicates that it is reasonable to obtain different results for different hypothesis spaces.**
> >
> > I fully appreciate the point made by the authors and don't think I am misunderstanding this. The R-divergence is expected to change if we consider a different architecture (hypothesis class) or loss functions and this property can be desirable.
> >
> > I would like to clarify if the experiments that evaluate the test power of R-div depend on the choice of architecture (hypothesis space) and the loss?  If this is case, can we expect R-divergence to be unable to distinguish different distributions (test power of 0.0) in some scenarios?  I don't see this as a drawback of the method but it would be helpful to discuss limitations.
> >
> >
> > **Q2: For both scenarios, R-divergence can quickly converge to a measurement representing the difference between two distributions as the sample size goes to infinity.**
> >
> > Proposition 4.3 doesn't talk about the rate of convergence and it would be incorrect to say that R-divergence converges **quickly**.
> >
> > **Q3: A perfect hypothesis from a rich class could obtain zero empirical risks on both datasets. In this case, this hypothesis for its learning task is not sensitive to the two distributions**
> >
> > I don't follow this argument. We can achieve zero empirical risk on the dataset $\hat p \cup \hat q$ but this does not imply that the hypothesis is not sensitive to $\hat q$.
> >
> > **Q3: The following experiment results show that R-divergence can easily separate samples with different classes from MNIST.**
> >
> > Thanks for running this experiment! I find it very interesting that cross-entropy losses differ between the classes even on MNIST.  Different classes seem to have different average losses which makes the permutation test work well in practice.
> >
> > I find the experiments compelling and have increased my score. I'd be be willing to increase it further if the authors could address some of the above questions.

---

> > > ### Author Response · Authors · 2023-08-13
> > > **Response to Reviewer 3YPD**
> > >
> > > Thank you very much for your recognition of our work. We appreciate your thoughtful feedback and have addressed each of the new questions you raised.
> > >
> > > **Q1. Name:**
> > >
> > > We deeply appreciate your meticulous research approach and constructive insight. Indeed, the R-Div can be regarded as a type of divergence. As indicated in the references you provided, R-Div satisfies non-negativity and non-negativity properties. It's important to note that the requirement of triangle inequality applies to metrics rather than divergences, and both KL divergence and R-Div do not satisfy this property.
> > >
> > > Your observations and feedback greatly contribute to the refinement of our work. In this revision, we have clarified that R-Div does not aim to provide a new definition of divergence. Following the definition of divergences, R-Div aims to estimate the discrepancy between two distributions for their given datasets. R-Div is designed according to a more reasonable insight than H-Div that two distributions are likely identical if the optimal hypothesis has the same expected risk on each distribution.
> > >
> > > Permutation test [R1] is widely applied to guarantee the significant level [R2], which indicates the probability of incorrectly deciding two different distributions. The permutation test is not a component of our proposed method. Instead, we adopt the permutation test to calculate the significant level and evaluate the performance of all the considered methods in experiments.
> > >
> > > **Q3. Scenario:**
> > >
> > > We sincerely apologize for any confusion caused. According to Theorem 4.1, R-Div is a general divergence measure that can be applied to various learning tasks.
> > >
> > > Specifically, given two datasets, it would be possible to train different learning tasks on them. The distinct learning tasks would have their respective loss functions, hypothesis spaces, and target functions. R-Div can be applied to estimate the discrepancy between the two datasets under any learning task. For example, R-Div can estimate the discrepancy for either the unsupervised reconstruction task with VAE and binary cross-entropy loss or for the supervised classification task with LeNet and cross-entropy loss.
> > >
> > > For the case study of learning with noisy labels, R-Div can distinguish clean and noisy samples by enlarging the discrepancy. Accordingly, we can train a robust network on clean samples and alleviate the misleading of noisy samples.
> > >
> > > **Q2. model-oriented:**
> > >
> > > Given two datasets, the R-Div values would change for different learning tasks or models. Accordingly, the discrepancy estimation performance depends on the hypothesis spaces, loss functions, and target functions of downstream tasks. This is reasonable because different learning tasks have different sensitivities to distribution discrepancies, which would be sensitive to task settings. Therefore, given two datasets, if we aim to quantify their discrepancy, it would be essential to specify the tasks to be applied on. R-Div can then calculate the discrepancy between these two datasets under the context of the specified task, i.e., with its hypothesis space, loss function, and target function.
> > >
> > > A zero-test power signifies that the algorithm is unable to distinguish between two distinct distributions. This has been verified by the results of our experiments in Figures 3, 4, and 5. This is not a drawback of R-Div. Instead, it is primarily attributed to the limited sample size within the dataset, which causes R-Div to incorrectly learn a minimum hypothesis over-fitting limited training samples across both datasets.
> > >
> > > We greatly appreciate your observation and will incorporate a comprehensive discussion on this issue in the final version.
> > >
> > > **Q2. Convergence:**
> > >
> > > We are deeply grateful for bringing the oversight to our attention. Allow us to reiterate the findings. Proposition 4.3 outlines that R-Div can quickly converge at the rate of $O(\sqrt{N})$. Corollary 4.5 indicates that our algorithm converges to the sum of two terms. Both can effectively quantify the discrepancy between datasets.
> > >
> > > **Q3. Hypothesis:**
> > >
> > > Given diverse hypothesis spaces, R-Div would yield different outcomes. A rich hypothesis space might exist for distributions with significantly distinct mathematical expressions, leading to the empirical risks of the minimum hypothesis on mixed data being identical for both datasets.
> > >
> > > If these empirical risks are still equal to zero for sufficient sample size, it essentially indicates that the model fails to differentiate between the two distributions. Thus, within this hypothesis space, the samples from the two given datasets can be treated as drawn from the same mixture distribution.
> > >
> > > In summary, we hope we have addressed your new comments and questions. We enjoy the constructive technical discussion with you, and many thanks for your valuable comments and discussion. Please let us know if you have any other comments or questions.

---

> > > > ### Comment · Reviewer_3ypD · 2023-08-14
> > > > **Thank you for the clarifying responses!**
> > > >
> > > > Thanks for all the clarifying responses.
> > > >
> > > > I thank the authors for pointing out that R-divergence is indeed a divergence. I was incorrectly thinking about the definition of a metric instead of a divergence. R-divergence doesn't strictly satisfy positivity ($D_R(p||q) = 0$ *iff* $p=q$), but that is not an issue since it is non-negative.
> > > >
> > > > I also thank the authors for clarifying that R-divergence works across a range of scenarios but the test power can sometimes be 0 when we have very samples. This is interesting, and I don't view this as a drawback since no statistic can be expected to have a good test power in all scenarios. I still wonder if R-divergence can also have problems with overfitting if we consider a very rich hypothesis space.
> > > >
> > > > I have raised my score since all my major concerns have been addressed.

---

> > > > > ### Author Response · Authors · 2023-08-14
> > > > > **Thank you for the improved score!**
> > > > >
> > > > > We genuinely appreciate your positive assessment of our research and the improved score. Regarding the two specific queries you raised, we offer the following responses:
> > > > >
> > > > >
> > > > > **Q1. positivity:**
> > > > >
> > > > >
> > > > > Your insights reflect a deep understanding of our method. Based on Eq. (3), we can deduce that theoretically, $ \mathcal {D} _ {R} (p \| q) $ iff $p = q$. However, it's important to note that we typically do not have access to the exact expressions of these two distributions. We apply an empirical estimator $\mathcal{\widehat{D}}_{\text{R}}(\widehat{p} \| \widehat{q})$ to approximate the divergence solely based on limited observed data, which inevitably lead to certain biases. On a theoretical note, we have also demonstrated that with an infinite sample size, the estimator can converge to this divergence, as shown in Corollary 4.4.
> > > > >
> > > > >
> > > > > **Q2. Overfitting:**
> > > > >
> > > > >
> > > > > This is a truly intriguing question. R-Div aims to address the over-fitting issue caused by learning a hypothesis and evaluating its empirical risk on the same dataset. However, within a rich hypothesis space, the issue of over-fitting persists. It is important to consider this from two distinct angles. In scenarios involving a limited sample within a rich hypothesis space, learning a perfect model fitting samples from both datasets is plausible. This over-fitting phenomenon could then result in an inability to differentiate between distinct distributions.
> > > > >
> > > > > Conversely, for a specific learning task with a substantial sample size, if there persists a minimum hypothesis from a rich hypothesis space that consistently yields 0 empirical risks for both datasets, we do not consider this as over-fitting. Instead, it indicates that within this specific task, the hypothesis space exhibits insensitivity to the discrepancies between the two datasets. Your in-depth comprehension of our work is truly appreciated, and we are committed to enhancing our discussion based on your insights in the revised version. Thank you for your valuable engagement.
> > > > >
> > > > >
> > > > > Once again, we express our gratitude, and if you have any further questions or suggestions, please feel free to let us know.

---

### Official Review · Reviewer_b1Kb · 2023-07-08

**Soundness:** 3 good
**Presentation:** 3 good
**Contribution:** 2 fair
**Rating:** 5
**Confidence:** 4

**Summary:**

This paper proposes R-divergence to measure the distributional discrepancy of two datasets. The general idea is to learn a minimum hypothesis on the mixed data and then define R-divergence as the difference of the empirical risk of the learned hypothesis between two datasets. The proposed divergence leverages the merits of both H-divergence and L-divergence. Sufficient experiments are performed to demonstrate the superior performance. The case study on learning with label noise is also interesting.

**Strengths:**

1. The method is very easy to follow. Authors also provide valid theoretical analysis (although some results rely heavily on previous literature).
2. Sufficient experiments in different learning scenarios (e.g., multiple domains, spurious correlations, and noise labels) are performed.
3. The case study on learning with label noises is interesting. However, a suitable comparison with state-of-the-art methods in this category can enhance the quality of this section.

**Weaknesses:**

1. It seems to me that R-divergence simply combines the merits of L-divergence and the recently proposed H-divergence. The original L-divergence is intractable since it requires evaluation of all hypotheses in the hypothesis space. By contrast, R-divergence just learns a minimum hypothesis on the mixed data. However, the general idea of defining divergence as the difference of risks on the mixed data and individual datasets is investigated in the H-divergence.
2. Since R-divergence is model-oriented. How to select hypothesis space and loss functions in practice?
3. The divergence or distance of distributions is an intrinsic property of two datasets. I can understand that a learning model-based approach provides a new solution. However, the examples in second paragraph of Introduction are hard to convince me why it is more meaningful to measuring divergence by considering a specific model; or why we really need a model-oriented discrepancy measure.

**Questions:**

Please see weaknesses points 2 and 3. Additionally, according to Table 1, it seems R-div is much better than others in case of limited samples. Can you give more analysis on the reason?

**Limitations:**

Authors did not discuss potential limitations or potential negative societal impacts. It would be better if authors can discuss How to select hypothesis space and loss functions in practice?

---

> ### Author Rebuttal · Authors · 2023-08-05
>
> We are grateful for your comments, below we address all and also clarify potential misreading and misunderstanding.
>
> **W1. Significance of R-divergence:**
>
> R-divergence is significantly different from L-  and H-divergence, discussed in the last paragraph of Introduction and Table 6. R- and L-divergences consider different hypotheses and for different scenarios. R- and H-divergences have different assumptions about the discrepancy evaluation principles and empirical risk calculation approaches. Specifically, R-divergence significantly improves performance by addressing the over-fitting issues of H-divergence.
>
> H-divergence assumes two distributions are different if the optimal decision loss is higher on their mixture distribution than on the individual ones. Accordingly, H-divergence trains a minimum hypothesis and evaluates its empirical risk on the same dataset, which causes small empirical risks for different datasets. The small empirical risks lead to a slight estimated discrepancy for two significantly different datasets, which fails to evaluate the discrepancy. In contrast, R-divergence addresses this issue by assuming that two distributions are likely identical if the optimal hypothesis has the same expected risk on each distribution. Accordingly, R-divergence merely learns a minimum hypothesis on the mixed data and evaluates its empirical risks on two individual datasets. R-divergence treats the empirical risk gap as the discrepancy, which ensures small and large gaps for similar and different datasets, respectively.
>
> Technically, R-divergence merely explores a minimum hypothesis on the mixed data, while L-divergence explores all the hypothesis space. Hence, R-divergence is model-oriented, considering their hypothesis space, loss function, target function, and the optimization process. Theoretically, R-divergence can be bounded by L-divergence, as shown in Corollary 4.5 and quickly converges for small L-divergence , as shown in Corollary 4.4. However, H-divergence cannot guarantee this property theoretically.
>
> **W2. Significance of model-oriented discrepancy:**
>
> It is essential to consider a model-oriented discrepancy measure because whether the samples of two datasets are drawn from the same distribution is relative. Specifically, different tasks have different requirements with corresponding hypotheses, loss functions, target functions, and optimization algorithms, inducing different sensitivities to distribution discrepancy.
>
> For example, the samples of the two datasets are drawn from different distributions for a binary classification task, while they tend to be treated as samples drawn from a mixture distribution for outlier detection and one-class classification tasks. Another example is that images of ‘Shiba Inu’ and ‘Corgi’ tend to be treated as samples from two different distributions if we aim to learn a binary classifier to distinguish them. However, they tend to be treated as sampled from a mixture distribution if we annotate both as ‘Dog’ and learn a multi-class classifier with ‘Cat’ images for a binary classifier to distinguish between ‘Dog’ and ‘Cat’.
>
> **W3. Hypothesis space and loss functions:**
>
> We believe there was a misreading of the definition. As stated in the second paragraph of Introduction, R-divergence is model-oriented for a specific learning model rather than hypothesis/loss function driven. Accordingly, it is unnecessary to select hypothesis spaces and loss functions for R-divergence, and R-divergence estimates the discrepancy for a specific learning task. Therefore, it is reasonable to obtain different results for different learning tasks.
>
> We describe that whether the samples from two datasets are drawn from the same distribution depends on a specific learning model. Different learning models hold different hypothesis spaces, loss functions, target functions, and optimization processes, thus having different sensitivity levels to distribution discrepancies. Therefore, the distributions of two given datasets can be treated as identical for some models but significantly different for others. Accordingly, in our experiments, we evaluate the discrepancy estimation performance for different network architectures (Table 5), datasets (Table 4), and tasks (Table 7 for reconstruction and Table 2 for classification).
>
> **Q1. Analysis for improved performance:**
>
> The discrepancy is distinct for two different distributions, even if the sample size is small, which benefits from addressing the over-fitting issue by learning the minimum hypothesis and evaluating its empirical risks on different datasets.
>
> R-divergence learns a minimum hypothesis on the mixed data. It achieves the same empirical risks on the two datasets where the samples are drawn from the same distribution. Contrarily, the empirical risks are different if the samples of two datasets are drawn from different distributions because the training dataset of the minimum hypothesis is not exactly the same as the two individual ones.
>
> From the theoretical perspective, as shown in Corollary 4.5, the empirical estimator of R-divergence can be bounded by discrepancy and L-divergence for significantly different distributions, which indicates that R-divergence can achieve a more distinct estimated discrepancy for different distributions. Please also refer to our response to Q1 for more information.
>
> **Q2. Case study:**
>
> The application of H-divergence in learning with noisy labels struggles to outperform the state-of-the-art approaches. This is because the research field has evolved over many years with significantly better results. The H-divergence application is more of an exploratory attempt, where we have incorporated H-divergence as a plugin within the baseline. Our work identifies a new avenue in this research field beyond establishing superiority. Furthermore, the case study illustrates the applicability of our proposed model in other domains.
>
> We will clarify the above in the final version.

---

> > ### Comment · Reviewer_b1Kb · 2023-08-17
> >
> > Thanks for your clarification. Many of my concerns are addressed.

---

> > > ### Author Response · Authors · 2023-08-17
> > > **Thank you for the raised score！**
> > >
> > > Thank you for taking the time to read our rebuttal. If you could kindly elaborate on specific points that you believe require additional attention or revision, we would be more than happy to address these in the revised manuscript.

---

> > > > ### Comment · Reviewer_b1Kb · 2023-08-20
> > > >
> > > > After reading the rebuttal between authors and Reviewer 3ypD, I do not have major concerns regarding this paper. R-divergence is model-oriented, which is actualy a "relative" concept and seems a bit different from previous literatures (especially those from Statistics) to rigorously perform two-sample test and quantify the distance with more enjoyable properties (e.g., Triangle inequality). The example of ‘Shiba Inu’ and ‘Corgi’ makes sense to me (if there are some experimental support on this example, that would be more interesting). Regarding label noise experiment, I understand that it is hard to compete with modern approaches (especially those combined with sophisticated components or too many engineering strategies). However, I do encourage authors to compete with some baslines. If R-divergence provides a new way to this problem, comparisons with baselines could attract more attentions or follow-up work.

---

> > > > > ### Author Response · Authors · 2023-08-20
> > > > > **Response to Reviewer b1Kb**
> > > > >
> > > > > We sincerely appreciate your thoughtful review of our paper and the time you dedicated to assessing our rebuttal with Reviewer 3ypD. Your comments have provided us with valuable insights, and we are grateful for your positive assessment of our work. In response to the two suggestions you raised, we have provided the following rebuttal:
> > > > >
> > > > > **Q1. Example:** We are pleased that the 'Shiba Inu' and 'Corgi' example resonated with you, and we will certainly endeavor to provide experimental support. Regrettably, due to the time constraints of the rebuttal process, we are unable to present experimental results within a short timeframe. However, we invite you to refer to our response to Q3 of Reviewer 3ypD and Table 7. We can observe different discrepancy estimation results for the MNIST dataset under supervised classification and unsupervised reconstruction tasks, which also verifies the significance of model-oriented discrepancy.
> > > > >
> > > > > **Q2. Label Noise Experiment:** We greatly appreciate your insights and suggestions. We have indeed conducted some comparison results with two baselines from the community of learning with noisy labels, including MentorNet [R1] and Co-teaching [R2]. We have presented some of our results as a preliminary comparison and will continue refining our experiments to include more comprehensive contrastive outcomes in the revised version. We are pleased that you understand our algorithm is hard to compete with more specialized approaches tailored for the label noise setting. As you rightly noted, our algorithm is a modest attempt and not intended to supersede existing state-of-the-art methods. We remain committed to adapting the proposed R-divergence to this label noise setting and expanding the applications to other research areas involving non-distributed data, including out-of-distribution detection, domain adaptation, and federated learning.
> > > > >
> > > > > Classification accuracy on CIFAR10 with pair and symmetry flipping.
> > > > >
> > > > > | Method    | Pretrained | MentorNet | Co-teaching | R-div |
> > > > > |------------|-------|-------|-------|-------|
> > > > > | Pair (40%)	| 67.8 | 69.36 | 72.48  | 68.51 |
> > > > > | Symmetry (50%)   | 66.05 | 71.10 | 74.02 | 67.71 |
> > > > >
> > > > >
> > > > > Once again, we express our gratitude for your constructive evaluation and encouragement. We look forward to your continued guidance and the opportunity to further elevate the standard of our work.
> > > > >
> > > > > [R1] Jiang et al., MentorNet: Learning Data-Driven Curriculum for Very Deep Neural Networks on Corrupted Labels. ICML, 2018.
> > > > >
> > > > > [R2] Han et al., Co-teaching: Robust training of deep neural networks with extremely noisy labels, NeurIPS, 2018.

---

### Official Review · Reviewer_EFgW · 2023-07-19

**Soundness:** 3 good
**Presentation:** 3 good
**Contribution:** 3 good
**Rating:** 7
**Confidence:** 3

**Summary:**

A new distance for comparing two samples of two distributuions p and q is proposed that is based on learning a model on the mixtre distribution of p and q. More precisely, the distance is the difference between errors of the learned model evaluated on the individual distributions. Theoretical statements on the estimation quality are given together with empirical evaluations.

**Strengths:**

- Simple method.
- Empirical results are very good.
- I could not detect any errors while checking the proofs.

**Weaknesses:**

- Theoretical upper bound on estimation quality depends on L-divergence between distributions. That is, if $\mathcal{D}_L(p||q)$ is large, the estimate of the porposed distance might have a very low quality even if the sample size goes to infinity.
- Citations are sometimes confusing, which makes the main argumentation hard to follow. For example, I cannot find a discussion of an "overfitting problem of H-divergence" in [50], as it is noted in the introduction. I also cannot find the reference [63] in [50], although [63] is given as reference of the H-divergence in the paper under review.
- Theoretical notation/assumptions could be extended, e.g.: What is $\mathcal{A}$? Does it need to have specific properties so that Theorem 1 is satisfied? What are the properties of the input/output spaces so that the thoerems hold? How do you exactly specify a learning model $\mathcal{T}$?

**Questions:**

- Do the authors have an idea about the convergence of the empirical estimator towards the true distance for increasing sample size, in the case when the L-distance is large? E.g. is there a lower bound on the left-hand side of Corollary 4.4?
- What is the effect of regularization of the trained neural networks (e.g., weight decay) on the porposed distance in the experiments?
- Concerning the main motivation of improving overfitting, is there an empirical evaluation comparing the "amount of overfitting" (however this is measured) of H-divergence and L-divergence?


**Limitations:**

- What is a possible effect in the test error, when the upper bounds in the theoretical statements are large, as e.g. in the case of large L-divergence?

---

> ### Author Rebuttal · Authors · 2023-08-05
>
> We thank your constructive comments.
>
> **W1. Theoretical upper bound:**
>
> We agree that a large L-divergence $\mathcal{D}_{\textup{L}} (p || q)$ will affect the convergence of the empirical estimator. However, a large L-divergence contributes to identifying whether samples of two datasets are drawn from different distributions as it indicates that the two distributions are significantly different. Accordingly, the empirical estimator will converge to the sum of the discrepancy and the L-divergence as the sample size increases, as shown in Corollary 4.5. Both discrepancy and L-divergence represent large distribution discrepancies. Hence, an empirical estimator represents the significant difference between two distributions. Conversely, a small L-divergence indicates that the empirical estimator rapidly converges to a small discrepancy for increasing the sample size, as shown in Corollary 4.4. Therefore, the empirical estimator also represents the small distribution discrepancy if the samples of two given datasets are drawn from the same distribution.
>
> **W2. Citations:**
>
> The network over-fitting issue is mentioned in [50], and H-divergence is proposed by [63]. Furthermore, in the last paragraph of Introduction on page 2, we claim that the over-fitting issues will cause a slight discrepancy estimated by H-divergence even if the two distributions are significantly different. This is because the datasets used to learn the minimum hypothesis and calculate their empirical risks are the same, which causes two similar and small empirical risks for the two distributions. Accordingly, the empirical risk gap is small for two significantly different distributions. The proposed R-divergence aims to address their over-fitting issues in estimating discrepancy by learning the minimum hypothesis and evaluating its empirical risks on different datasets.
>
> **W3. Theoretical notation:**
>
> In the second paragraph of Section 2 on page 2, we state that $a \in \mathcal{A}$ is the target function, e.g., $a(x) = y$ for supervised classification and $a(x) = x$ for unsupervised input reconstruction. The theorems reflect general results for any input and output spaces. In the second paragraph of Introduction on page 1, we describe that whether the samples from two datasets are drawn from the same distribution depends on a specific learning model. The distributions of two given datasets can be treated as identical for some models but significantly different for others. Different learning models hold different hypothesis spaces, loss functions, target functions, and optimization processes, thus having different sensitivities to distribution discrepancies. For example, ‘apple’ and ‘banana’ are significantly different for a binary classification task. However, if our learning task is to learn a binary classifier to distinguish between ‘animal’ and ‘fruit’, the binary classifier can treat ‘apple’ and ‘banana’ as being drawn from the same mixture distribution and is not sensitive to their discrepancy. Accordingly, we propose the model-oriented distribution discrepancy evaluation. Therefore, instead of selecting a specific learning model $\mathcal{T}$, the theory of R-divergence evaluates the discrepancy between two datasets for the model.
>
> **Q1. Theoretical lower bound:**
>
> We are usually not concerned about the lower bound of the convergence because a zero lower bound can ensure that the empirical estimator tends to the discrepancy for both small and large $\mathcal{D}_{\textup{L}} (p \| q)$. The upper bound is more essential to analyze whether the empirical estimator can represent the difference between two datasets. As discussed in your Q1, the upper bound is small for a small discrepancy, which indicates that the empirical estimator tends to the discrepancy. Furthermore, the upper bound is large for a large discrepancy. However, the empirical estimator converges to the sum of discrepancy and the large L-divergence, where both of them can represent the difference between two datasets, as shown in Corollary 4.5.
>
> **Q2. Effect of regularization:**
>
> Regularization is not significant for discrepancy estimation methods. We evaluate the discrepancies with and without weight decay (WD) and obtained the same results. The regularization aims to alleviate the over-fitting issue for improving the generalization. The regularization could increase the training losses. However, it still causes similar empirical risks on different training datasets, which causes a slight empirical risk gap between two significantly different datasets for the considered discrepancy estimation task. Therefore, the network regularization cannot address the over-fitting issue and make the discrepancy more distinct for different distributions and datasets.
>
> | Datasets| A+C | A+P | A+S | C+P | C+S | P+S | Ave. |
> |------------|-------|-------|-------|-------|-------|-------|-------|
> | R-Div w/ WD| 1.000 |1.000 |1.000 |0.524 |0.572 |0.978 |0.846 |
> | R-Div w/o WD| 1.000 |1.000 |1.000 |0527 |0589 |0980 |0.849 |
>
> **Q3. Amount of over-fitting:**
>
> The amount of over-fitting, i.e., the empirical risk, for L-divergence, H-divergence, and R-divergence on MNIST (N = 200) are reported below. The results show that R-divergence can obtain a much more distinct discrepancy due to the significantly different empirical risk values for $p$ and $q$. This is because R-divergence addresses the over-fitting issue by optimizing a minimum hypothesis on a mixed dataset and evaluating its empirical risks on different datasets. Therefore, R-divergence can induce a more distinct discrepancy for two significantly different distributions.
>
> | Methods | Empirical Risk for p  | Empirical Risk for q   | Estimated Discrepancy |
> |------------|-------|-------|-------|
> | L-Div    | 713.0658 | 713.1922 | 0.0212 |
> | H-Div    | 111.7207 | 118.9020 | 1.6603 |
> | R-Div    | 144.0869 | 99.6166 | 44.4703 |
>
> We will clarify the above in the final version.

---

> > ### Comment · Reviewer_EFgW · 2023-08-16
> > **Thank you for your answer**
> >
> > I appreciate the answers of the reviewers. My suggestion remains to accept the paper.

---

> > > ### Author Response · Authors · 2023-08-16
> > > **Thank you for your acceptance!**
> > >
> > > We sincerely appreciate your acknowledgment and acceptance of our paper. Your engagement with our paper has been both insightful and motivating. We look forward to any additional feedback you may provide and the opportunity to enhance our work based on your valuable input.

---

### Author Rebuttal · Authors · 2023-08-05

We are grateful for all the reviewers for their valuable feedback and insightful comments on our paper.

We are heartened to see that two of the reviewers have acknowledged the significance, contribution and quality of our work, where both assign a score of 7. We appreciate their positive remarks and thank them for acknowledging the significance of our research.

However, we acknowledge that the reviews also demonstrate a notable divergence of perspectives among the reviewers. We are also serious about the divided comments made by two other reviewers. After carefully reading comments by Reviewer b1Kb, s/he misreads or misunderstands our work by mixing our proposed R-divergence with a simple combination of L-divergence and H-divergence, which is incorrect and inconsistent with our design and statement in the paper. Accordingly, s/he misunderstands that our method is sensitive to hypothesis space and loss function, which is again incorrect and inconsistent with our design. We clarify their misunderstanding in the rebuttal.

Further, Reviewer 3ypD basically misunderstands our work perhaps also because of lacky of background expertise in the area. His/her questions do not conform to our motivation of proposing R-divergence to address the over-fitting issue relating to a specific learning task. Their misunderstandings with the corresponding inappropriate examples may be because of their misalignment of our work to the related work on L-divergence and H-divergence. We clarify their misunderstandings in the rebuttal.

Triggered by the notable divergence between the reviewers, during the rebuttal, we treat each comment seriously, especially those misunderstandings, and have made significant changes to address all comments and substantially improved the paper quality. We have provided more detailed explanations to address some of the points relating to misreadings and misunderstandings of our research setup and motivation. We are confident that the improved manuscript will be better aligned with the expectations of all the reviewers. In short, the main changes are summarized below. We have:

1. provided more in-depth analyses of the theoretical results in order to explain why the algorithm converges rapidly and achieves favorable outcomes;

2. adjusted the citations to avoid causing confusion and misunderstandings;

3. offered additional examples and analyses to illustrate the significance of model-oriented discrepancy estimation;

4. present the amount of over-fitting to validate the motivation;

5. provided further explanations to illustrate the distinctions between the proposed approach and existing methods;

6. provided additional analyses to explain why the proposed method can achieve impressive results even with limited samples;

7. provided more explanations for the failure cases and performed extra experiments with different network architectures on different tasks to verify the effectiveness of the proposed methods;

8. added experiments to verify R-divergence is not sensitive to data scales;

9. performed extra experiments on different corruption types and levels to verify the effectiveness of the proposed methods.

Once again, we extend our sincere appreciation to all reviewers for their time and effort in evaluating our work. We are committed to enhancing the quality of our research and making it valuable to the community by addressing all valuable and constructive comments. Please feel free to reach out if there are any further questions or clarifications needed. We are eagerly looking forward to the opportunity to present our improved work to you.

---

### Decision · Program_Chairs · 2023-09-21

**Decision:**

Accept (poster)

**Comment:**

This paper introduces a novel method for quantifying the divergence between two probability distributions. Reviewers have recognized the significance of this contribution, highlighting the simplicity and strong theoretical foundation of the presented method. Furthermore, the experimental results have been found to be compelling. One minor concern, raised by a reviewer, pertains to the lack of clarity regarding why the proposed R-divergence appears to be unaffected by overfitting while the H-divergence is susceptible to it. It is advisable for the authors to provide additional clarification or discussion on this matter in the final version of the paper.